# DENOISING GRAPH DISSIPATION MODEL IMPROVES GRAPH REPRESENTATION LEARNING

## ABSTRACT

Graph-structured data are considered non-Euclidean as they provide superior representations of complex relations or interdependency. Many variants of graph neural networks (GNNs) have emerged for graph representation learning which is essentially equivalent to node feature embedding, since an instance in graph-structured data is an individual node. GNNs obtain node feature embedding with a given graph structure, however, graph representation learning tasks entail underlying factors such as homophilous relation for node classification or structure-based heuristics for link prediction. Existing graph representation learning models have been primarily developed toward focusing on task-specific factors rather than generalizing the underlying factors. We introduce Graph dissipation model that captures latent factors for any given downstream task. Graph dissipation model leverages Laplacian smoothing and subgraph sampling as a noise source in the forward diffusion process, and then learns the latent factors by capturing the intrinsic data distribution within graph structure in the denoising process. We demonstrate the effectiveness of our proposed model in two distinct graph representation learning tasks: link prediction tasks and node classification tasks, highlighting its capability to capture the underlying representational factors in various graph-related tasks.

## 1 INTRODUCTION

A fundamental concept in representation learning is that data distributions have effective lower-dimensional structures. For example, consider image data, which is presumed to exist on a lower-dimensional manifold within the pixel-space. This assumption relies on the presence of a collection of underlying factors that capture the semantics of an image. However, graph-structured data are considered non-Euclidean since they represent complex interdependency or relations that extensively exist in networks, *e.g.,* citation network, social network, interaction network, and neuron connectome.

Since data instances in a network graph are individual nodes, graph representation learning essentially reduces to learning node embeddings. Thus, graph representation learning has evolved predominantly with node classification tasks and graph classification tasks. There has been growing attention on link prediction tasks recently, however, models that perform well in node classification tasks do not necessarily promise a similar level of performance in link prediction tasks. This disparity results from the unique characteristics of link prediction tasks that edges form based not only on node feature embeddings but also on structure-based information such as neighborhood-overlap heuristics or higher-order heuristics. Existing graph representation models such as Graph neural networks(GNNs), which heavily rely on node feature embeddings, often struggle to effectively capture some structural information that is required for more accurate link prediction. In this manner, underlying latent factors of a network graph required for learning optimal representations vary depending on the specifics of graph representation learning tasks. Still, graph representation learning models are not capable of learning latent factors of network graphs without explicit task-oriented assumptions.

This work aims to capture the comprehensive and integrated latent factors of a graph that are not limited to a specific downstream task. However, the challenge of learning latent factors of a graph is that it is difficult to define it within a family of known probability distributions since arbitrary

underlying structures are complex but unknown, *i.e.,* non-Euclidean. This problem becomes more challenging in network graphs. A network graph constitutes an entire data, and it lacks well-defined rules or assumptions regarding the optimal results.

We introduce Graph dissipation model (GDM) based on a diffusion model, which learns the comprehensive latent distribution of the graph, enabling it to effectively solve any given downstream tasks without task-specific assumptions. Graph dissipation model captures the latent factors of a network graph, owing to its diffusion model architecture with the intuition of capturing arbitrary data distribution. Our model, GDM, has novel approaches. GDM leverages Laplacian smoothing as a noise source of the feature diffusion process, incorporating *over-smoothing* and the concept of *dissipation*. We encourage node features in a graph to be smoothed (*i.e.,* blurred) by Laplacian smoothing based on Laplacian matrix since it preserves inherent structural characteristics of a network graph, *i.e.,* node dependency. Besides, Laplacian smoothing is a particular case of diffusion process across a graph, where information flows between neighboring nodes. This interpretation aligns with dissipation-based diffusion models (*e.g.,* Rissanen et al. (2022)). We exploit the intuition that information or *signal* is not only smoothed but also erased as it flows between instances (i.e., nodes) within graph structures, leading to our unique approach of utilizing *over-smoothing* as the final state of the feature diffusion process. Namely, there are dissipation of signal while feature information of a graph is blurred through iterative Laplacian smoothing during the diffusion process from GDM. Lastly, GDM conveys signal dissipation from feature space to a graph structure by defining Dissipative structure sampling, a subgraph sampling that reflects feature dissipation, in the structural diffusion process. Our objective is capturing latent factors underlying a network graph, leading to optimal representations applicable to various graph representation learning tasks while naturally regarding specifics inherent in a given task, e.g., node classification or link prediction. GDM is a diffusion model-based graph representation learning model that is universally applicable to network graph representation learning tasks without explicit task-oriented assumptions. The contributions of the paper are summarized as follows:

- We propose Graph dissipation model (GDM) leverages the intuition from diffusion models to address the motivation that underlying latent factors of a network graph are complex but unknown, which leads graph representation learning to relying on task-oriented approaches. To the best of our knowledge, GDM is the initial work on network graph representation learning that raises and addresses such motivation.

- GDM introduces a unique perspective by defining Laplacian smoothing as a noise source and over-smoothing as a convergence state. Theoretically, Laplacian smoothing as a noise source of a diffusion model aligns with the intuition of diffusion models in image domain, especially in resoultion perspective. Also, we leverage feature-based structure sampling to lift dissipation in features to a graph structure during the structural diffusion process.

- We demonstrate the effectiveness of GDM in two downstream tasks, link prediction and node classification tasks on 7 benchmark datasets. In addition, we conduct ablation studies to provide insights into which component is advantageous to the given task.

## 2 RELATED WORK

**Denoising Diffusion Probabilistic Models.** Denoising diffusion probabilistic models (DDPMs), or diffusion models, have become powerful generative models in computer vision tasks. Sohl-Dickstein et al. (2015) proposed a deep unsupervised learning framework, known as Diffusion probabilistic models, based on nonequilibrium thermodynamics. Closely related to this, Ho et al. (2020) introduced Denoising Diffusion Probabilistic Models (DDPMs), the powerful generative model that gradually perturbs data with Gaussian noise in a diffusion process for learning probabilistic models, then learning data distribution by an iterative denoising process. Song et al. (2020) introduced a modified denoising diffusion process to non-Markovian diffusion process to accelerate efficiency. Rissanen et al. (2022) introduce a novel methodology parametrized by inverse heat equation instead of diffusion processes, reflecting multi-resolution inductive bias. Furthermore, DDPMs or diffusion models are not only used for generation tasks (Ho et al., 2020; Dockhorn et al., 2021; Bao et al., 2022) but also for other tasks. The latent representations obtained through diffusion models have been used for diverse computer vision tasks *e.g.,* image segmentation (Baranchuk et al., 2021) and image classification (Zimmermann et al., 2021).

**Graph Representation Learning.** As a data point in graph-structured data is a node, prevalent Graph neural networks usually demonstrate their efficacy on node classification tasks. GCN (Kipf & Welling, 2017) defines convolutional operation in graph domains to aggregate messages or information of neighboring nodes. This work emphasizes the semi-supervised node classification setting that is inherent in graph structures due to nodes' interdependency. GAT (Veličković et al., 2018) improves graph representation learning by allowing nodes to attend to each neighboring node with varying degrees of importance which is learned through attention mechanism. GRAND (Chamberlain et al., 2021) approaches graph representation learning as a continuous diffusion process that information or heat diffused on a graph, and interprets existing GNNs as discretizations of an underlying partial differential equation of graph diffusion. Unlike node classification tasks, link prediction tasks do not solely rely on node embedding. Zhang & Chen (2018) investigated the importance of structure-based heuristics in link prediction tasks and proposed SEAL that extracts $h$-hop enclosing subgraph to learn structural features to enhance link prediction tasks. On top of that, Neo-GNNs (Yun et al., 2021) and NBFNet (Zhu et al., 2021) generalize neighborhood overlap heuristics and Bellman-Ford Algorithms to capture useful structural information for link prediction tasks, respectively. However, existing graph representation learning models for network graphs focus only on either node classification tasks or link prediction tasks. Our work aims to improve both node classification tasks and link prediction tasks by leveraging insight from diffusion models.

**DDPMs on Graph domain.** In terms of generative graph models, Ma et al. (2019) and Elinas et al. (2020) introduce early variational methods to learn graph representation employing independent Bernoulli distribution as a graph distribution. Jo et al. (2022) proposed score-based generation model that learns joint distribution of nodes and edges. Vignac et al. (2022) adopted a diffusion model to generate molecular graphs, defined with categorical distribution. Haefeli et al. (2022) generates random graph structure and emphasizes graph domain benefits from discrete time-space than continuous time-space. Chen et al. (2023) propose an efficient graph generation methodology for generating large-scale random graphs by perturbing structures with an edge removal process that drops all the edges connected to selected nodes.

## 3 PRELIMINARY

### 3.1 LAPLACIAN SMOOTHING

The Laplacian smoothing operation in a graph is based on the Laplacian matrix, denoted by $L$, which captures the structural properties and propagates signal on a graph structure. According to Chung (1997), the unnormalized Laplacian matrix is defined as $L = D - A$, where $A$ is an adjacency matrix and $D$ is a degree matrix of $A$, i.e., $D = \text{diag}(d_1, d_2, ..., d_N), d_i = \sum_j \tilde{A}_{ij}$. Given an initial node feature matrix $X \in \mathbb{R}^{N \times F}$, the smoothed feature representation $X'$ obtained by Laplacian smoothing (Taubin, 1995), i.e., $x'_i = x_i + \lambda \triangle x_i$. $\triangle$ is a Laplacian operator and $\lambda$ is a scaling coefficient that controls the extent of the smoothing operation, i.e., $0 < \lambda \leq 1$. This can be rewritten in the matrix formulation as

$$X' = (I - \lambda D^{-\frac{1}{2}} L D^{-\frac{1}{2}})X = (I - \lambda L_{sym})X,$$
$$X' = (I - \lambda D^{-1} L)X = (I - \lambda L_{RW})X,$$

where $I$ denotes the identity matrix. Along with this, $L_{sym}$ and $L_{RW}$ indicate two variants of normalized Laplacian matrices. Laplacian smoothing produces the diffusion of signal across the graph, leading to a filtered representation of the signal on the graph structure with respect to neighborhood nodes' features. Note that Laplacian smoothing can be applied iteratively to propagate the signal on the graph further, gradually blurring node representations.

**Over-smoothing.** As the Laplacian smoothing operation is performed multiple times, the signal from neighboring nodes gets increasingly diffused, leading to a convergence of node representations towards a common average value (Oono & Suzuki, 2019; Keriven, 2022). This convergence eliminates the subtle differences between nodes, blurring out the important structural and contextual representation in the graph. Thus, the over-smoothing problem makes the node features indistinguishable. Theoretical proof of over-smoothing is in Appendix B.

## 3.2 Denoising Diffusion Probabilistic Model

Denoising Diffusion Probabilistic Models(DDPMs) or Diffusion models are defined by two processes: a forward process that gives discriminative noise on input images and a reverse process that learns data distribution by denoising tasks. Let a data instance be sampled from a real date distribution $\boldsymbol{x}_0 \sim p_{\text{data}}$, a forward diffusion process produces a sequence of noisy data samples $(\boldsymbol{x}_1, \boldsymbol{x}_2, ..., \boldsymbol{x}_T)$ by adding random Gaussian noise to the given data sample at time step $t$ with variance $\beta_t$ from variance schedule $\{\beta_t \in (0,1)\}_{t=1}^T$. The significance of diffusion models is that a forward diffusion process is a Markov chain that gradually adds Gaussian noise, thus, the posterior distribution $q(\boldsymbol{x}_{1:T}|\boldsymbol{x}_0)$ is approximated under Markov property and variance schedule (Ho et al., 2020),

$$q(\boldsymbol{x}_{1:T}|\boldsymbol{x}_0) = \prod_{t=1}^T q(\boldsymbol{x}_t|\boldsymbol{x}_{t-1}),$$

$$q(\boldsymbol{x}_t|\boldsymbol{x}_{t-1}) := \mathcal{N}(\sqrt{1-\beta_t}\boldsymbol{x}_{t-1}, \beta_t\mathbf{I}).$$

$\beta_t$ can be held constant or learned by reparametrization trick, however, Ho et al. (2020) sets $\beta$ as hyperparameters. Hence, a forward diffusion process does not contain trainable parameters.

In a reverse denoising process, on the other hand, a denoising model $p_\theta$ learns to invert the noisy sequence obtained in the forward diffusion process. In a reverse denoising process, a denoising model would be able to regenerate the sample from a Gaussian noise input $\boldsymbol{x}_T \sim \mathcal{N}(\mathbf{0}, \mathbf{I})$ as it inverts the forward process, extracting the distribution $q(\boldsymbol{x}_{t-1}|\boldsymbol{x}_t)$. Since $q(\boldsymbol{x}_{t-1}|\boldsymbol{x}_t)$ is intractable, a denoising model $p_{[\theta]}$ approximate the distribution as follows:

$$p_\theta(\boldsymbol{x}_{t-1}|\boldsymbol{x}_t) := \mathcal{N}(\boldsymbol{x}_{t-1}; \mu_\theta(\boldsymbol{x}_t, t), \Sigma_\theta(\boldsymbol{x}_t, t)),$$

$$p_\theta(\boldsymbol{x}_{0:T}) = p(\boldsymbol{x}_T) \prod_{t=1}^T p_\theta(\boldsymbol{x}_{t-1}|\boldsymbol{x}_t).$$

Gaussian noise term $\mu_\theta$ is reparametrized to minimize the distance from $\mu_t$ which equals noise prediction. The intuition behind these processes is that trainable network $p_\theta$ learns an arbitrary data distribution by filtering out noise based on an assumed distribution $q$. To approximate the conditional probability distribution in the reverse process,

## 4 Graph Dissipation Model

Graph dissipation model (GDM) aims to learn latent representations from a graph that is universally applicable to various network graph representation learning tasks while naturally regarding specifics of those tasks without explicit task-oriented assumptions. Graph dissipation model (GDM) is a diffusion model framework for network graph representation learning. As illustrated in Fig.1, GDM consists of two parts, the forward process and the reverse process. To dissipate graph signals with the aspect of feature and structure simultaneously, we define Laplacian smoothing as noise source and propose dissipative structure sampling regarding dissipation. From spectral perspective, leveraging Laplacian smoothing gives promising support for capturing latent factors of network graph. During the reverse process, GDM learns the latent distribution with its own Denoising network $f_\theta$.

**Notations.** Consider an undirected graph $\mathcal{G} = (\mathcal{V}, \mathcal{E})$ with $N$ nodes, denoted by $\mathcal{V} = \{v_1, v_2, \ldots, v_N\}$, and a set of edges denoted by $\mathcal{E}$. The adjacency matrix $\boldsymbol{A} \in \mathbb{R}^{N \times N}$ is defined by $\boldsymbol{A}_{ij} = 1$ if $e_{ij} \in \mathcal{E}$ and 0 otherwise. Each node in $\mathcal{G}$ has a feature vector $\boldsymbol{x}_i \in \mathbb{R}^{1 \times d}$ of dimension $d$, and the collection of these feature vectors is represented by the matrix $\boldsymbol{X} \in \mathbb{R}^{N \times d}$, *i.e.*, $G = (\boldsymbol{A}, \boldsymbol{X})$.

## 4.1 Forward Process

To simultaneously blur and dissipate graph-structured data, we leverage a coupled diffusion process that merges feature space and structural space. Given the graph $G = (\boldsymbol{A}, \boldsymbol{X})$, the diffusion on the graph involves information dissipation, *i.e.,* frequency decay. We define the noise source of the forward process of GDM with Laplacian smoothing operation. According to Corollary B.1,

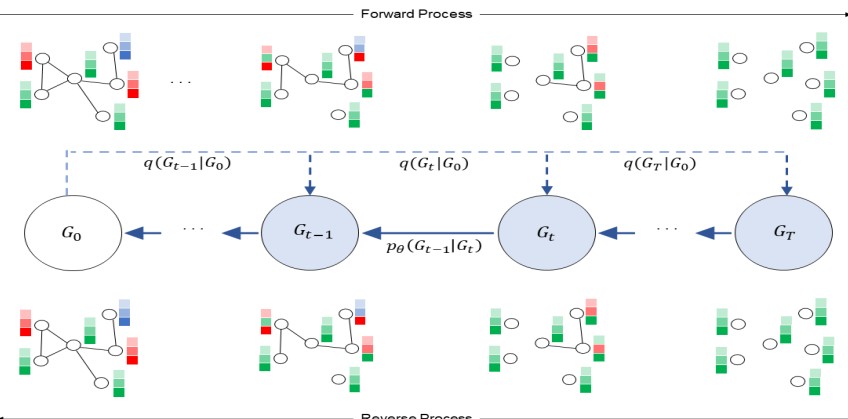

Figure 1: Graphical Model of Graph dissipation model. Our model leverages the Laplacian smoothing to define the forward process, inducing signal dissipation on a graph and reflecting the important aspect of a graph domain, i.e., node dependency. As Laplacian smoothing assures signal dissipation on feature space, GDM lifts dissipation from feature to a graph structure by *dissipative structure sampling*.

iterative Laplacian smoothing operation blurs out node features that converge to over-smoothing which makes each node indistinguishable. Laplacian smoothing directly operates on node features as a noise source. Smoothed blurry feature of Markov state $t$ is obtained as,

$$\boldsymbol{X}_t = (\mathbf{I} - \alpha \boldsymbol{L})\boldsymbol{X}_{t-1} = (\mathbf{I} - \alpha \boldsymbol{L})^t \boldsymbol{X}_0, \tag{1}$$

where $t$ denotes time step $t$ and $\boldsymbol{X}_0$ is an initial feature matrix.

Ultimately, Laplacian smoothing assures dissipation on a network graph. Laplacian smoothing bridges the gap between dissipation and graph representation. Since we can rewrite Laplacian smoothing using eigendecomposition, transforming to a spectral domain,

$$\boldsymbol{X}_t = (\mathbf{I} - \alpha \boldsymbol{L})^t \boldsymbol{X}_0 = U(\mathbf{I} - \alpha \Lambda)^t U^\top \boldsymbol{X}_0. \tag{2}$$

$U$ forms a basis for the graph spectral domain and the diagonal matrix $\Lambda$ contains the eigenvalues, which represent the frequencies corresponding to each eigenvector (Belkin & Niyogi, 2001). Specifically, $(\mathbf{I} - \alpha \Lambda)^t$ implies the decay of high frequencies on the graph spectral domain. As high-frequency components are decayed, the feature noise $(\mathbf{I} - \alpha \Lambda)$ converges towards a smooth signal that resides in the low-frequency components on the graph spectrum.

In other words, when high frequency gradually decays, the difference between signals also gradually diminishes in the spectral domain. In the spatial domain of a graph, it is interpreted as a loss of discrepancy in feature information among distinct nodes. This implies that the amount of decayed signal or information discrepancy varies for each node at each time step, converging over-smoothed feature. This aligns with the intuition of diffusion models, suggesting that our model GDM can learn the latent factors of a given graph by recovering this dissipated signal or information. Additionally, in real-world scenarios, as noise or missing information (e.g., missing links) exists in the features or adjacency of a network graph, the observations may not constitute perfect ground truth. This associates graph representation learning with inferring the most optimal graph information from a noisy observed graph. From the image-resolution perspective, our approach is also analogous to diffusion models utilizing a coarse-to-fine strategy to enhance resolution quality. This supports that our proposed approach shows promising results on capturing latent factors underlying a network graph, leading to optimal representations applicable to various graph representation learning tasks while naturally regarding specifics inherent in a given task.

Note that, our feature diffusion process can follow Markov chain property but also we can factorize Laplacian smoothing until time step $t$ based on Eq.1. Therefore, the feature diffusion process is written as

$$q(X_{1:T}|X_0) = \prod_{t=1}^{T} q(X_t|X_0), \quad q(X_t|X_0) := (\mathbf{I} - \boldsymbol{L})^t \boldsymbol{X}_0|\boldsymbol{X}_0, \tag{3}$$

letting $\alpha = 1$. We termed the decrease in the differences between node features as dissipation of signal. Signal dissipation is naturally defined in feature space, however, defining signal dissipation on graph structure is complicated to obtain directly. For a straightforward approach, we lift the dissipation of features to the graph structure. To lift feature dissipation to the graph structure, we define the structural diffusion process with *dissipative structure sampling* based on subgraph sampling as follows:

$$\hat{\boldsymbol{X}}_t = \boldsymbol{X}_t + \boldsymbol{\epsilon} \quad \text{where} \quad \boldsymbol{\epsilon} \sim \mathcal{N}(\mathbf{0}, \zeta\mathbf{I}) \tag{4}$$

$$\boldsymbol{A}_t[ij] \sim \text{Bern}(\boldsymbol{A}_t|\boldsymbol{A}_{t-1}[ij] = 1, p = s(\hat{\mathbf{x}}_i^{(t)}, \hat{\mathbf{x}}_j^{(t)})) \tag{5}$$

where $\zeta$ is a relaxation hyperparameter to prevent similarity converges to 1. $\hat{\boldsymbol{x}}_i^{(t-1)}$ denotes a feature vector of node $v_i$ at time step $t-1$ and $s, p$ denotes a similarity function and drop probability, respectively. The structural diffusion process follows Markov chain property, implying gradual dissipation of structural information reflecting dissipation of graph signals. The structure diffusion process is defined with Binomial distribution,

$$q(A_{1:T}|A_0) = \prod_{t=1}^{T} q(A_t|A_{t-1}), \quad q(A_t|A_{t-1}) := \mathcal{B}(\boldsymbol{A}_t|\boldsymbol{A}_{t-1}, s(\hat{\boldsymbol{X}}_t)). \tag{6}$$

Consequently, $q(X_{1:T}|X_0)$ and $q(A_{1:T}|A_0)$ can provide a broader range of underlying patterns as it increases data diversity, considering that a network graph is an entire dataset on its own.

## 4.2 REVERSE PROCESS

The reverse process $p_\theta$ models the posterior of the previous state given the current state. Let the forward process be $q(G_{1:T}|G_0)$ since it is a coupled process and the underlying pattern of a graph relies on both feature and structural representation. Then, we can optimize a denoising network $f_\theta$ by maximizing $\log p(G_0)$ as follows:

$$-\log p(G_0) \le \mathbb{E}_{q(G_{1:T}|G_0)}\left[-\log \frac{p_\theta(G_{0:T})}{q(G_{1:T}|G_0)}\right] \tag{7}$$

$$= \mathbb{E}_{q(G_{1:T}|G_0)}\left[-\log \frac{p(G_T)}{q(G_T|G_0)} - \sum_{t=2}^{T} \log \frac{p_\theta(G_{t-1}|G_t)}{q(G_{t-1}|G_0)} - \log p_\theta(G_0|G_1)\right] \tag{8}$$

The first term does not require learnable parameters since it is constant. However, the posterior of the forward process $q(G_{t-1}|G_t, G_0)$ has no closed-form expression. To approximate $q(G_{t-1}|G_t, G_0)$, we decompose $G$ into $X$ and $A$. Then, the loss function for GDM $\mathcal{L}_{\text{GDM}}$ is derived as follows:

$$\sum_{t=2}^{T} \mathbb{E}_q D\left[q(X_{t-1}|X_0)\|p_\theta(X_{t-1}|X_t)\right] + \sum_{t=2}^{T} \mathbb{E}_q D\left[q(A_{t-1}|A_0)\|p_\theta(A_{t-1}|A_t)\right]$$
$$+ \mathbb{E}_q\left[-\log p_\theta(X_0|X_1)\right] + \mathbb{E}_q\left[-\log p_\theta(A_0|A_1)\right] =: \mathcal{L}_{\text{GDM}} \tag{9}$$

According to Eq. 1, $D\left[q(X_{t-1}|X_0)\|p_\theta(X_{t-1}|X_t)\right]$ is equivalent to predicting less smooth features which means deblurring signal dissipation on feature space.

$$D\left[q(X_{t-1}|X_0)\|p_\theta(X_{t-1}|X_t)\right] = \|f_\theta(X_t, A_t) - X_{t-1}\|_2^2.$$

Since we lift feature dissipation to the forward structural process, under the mild assumption, $q(A_t|A_{t-1})$ can be approximated, *i.e.*, $q(A_t|A_{t-1}) \approx q(A_t|A_0)$. Note that, to make the graph structure sparser as the node features converge to oversmoothing, we defined the forward structural process with stochastic structure sampling dependent on features.

$$q(A_{ij}^{(t-1)}|A_{ij}^{(0)}) = \mathcal{B}(A_{ij}^{(t-1)}; p \overset{\propto}{\sim} LX = I - (I - LX)), \quad \text{if } A_{ij}^{(0)} = 1 \tag{10}$$

The edge probability $p$ estimation has uncertainty because we lift the feature distance upon edge existence probability through the forward structural process. However, due to the intuition of the forward structural process, edge probability $p$ is approximately correlated to Laplacian matrix which feature dissipation relies on. The intuition behind the forward structural process is lifting signal dissipation to a graph structure. Leveraging this intuition, the edge probability $p$ can be estimated

by discrepancy of structural information which implies dissipation on a graph structure. Therefore, $D\left[q(A_{t-1}|A_0)\|p_\theta(A_{t-1}|A_t)\right]$ is approximated with a discrepancy between $L$ and $L_{t-1}$,

$$D\left[q(A_{t-1}|A_0)\|p_\theta(A_{t-1}|A_t)\right] = \|f_\theta(X_t, A_t) - (L_0 - L_{t-1})\|_2^2$$

predicting the discrepancy between graph Laplacian where dissipation is dependent.

Therefore, the loss for Graph dissipation model is defined as

$$\mathcal{L}_{\text{GDM}} = \beta_t \underbrace{\sum_{t=2}^{T} \|f_\theta(X_t, A_t) - X_{t-1}\|_2^2}_{\mathcal{L}_{\text{feat}}} + \gamma \underbrace{\sum_{t=2}^{T} \|f_\theta(X_t, A_t) - (L_0 - L_{t-1})\|_2^2}_{} - \mathcal{L}_{\text{Lap}}$$

$$+ \beta_0 \underbrace{\|f_\theta(X_1, A_1) - X_0\|_2^2}_{\mathcal{L}_{\text{feat-recon}}} + \lambda \underbrace{\text{BCE}(f_\theta(X_1, A_1), A_0)}_{\mathcal{L}_{\text{recon}}}, \quad (11)$$

where $\beta_0, \gamma, \beta_1$ and $\lambda$ denotes weighting hyperparameters. Hyperparameter sensitivity analysis is in Appendix A.2. Finally, the total loss of Graph dissipation model can be written as follows:

$$\mathcal{L} = \mathcal{L}_{\text{GDM}} + \mathcal{L}_{\text{task}}, \quad (12)$$

where $\mathcal{L}_{\text{task}}$ is a downstream task loss.

Additionally, we design the architecture of the denoising network $f_\theta$ to effectively learn comprehensive latent distribution with aspects of both features and structures. Our denoising network $f_\theta$ consists of 2 layers of multilayer perception (MLP) as the encoder and 3 layers of MLP as the decoder for denoising tasks. Specifically, the decoder can be shared as the predictor when a downstream task handles link prediction tasks. Since the forward process in GDM converges to overly blurred features and nearly empty structures, we define the learnable parameters, latent Laplacian values in the denoising network, to incorporate the minimum latent information during the reverse process and stabilize the learning towards denoising tasks. We also define the predictor for each downstream task, link prediction task, and node classification task. For the link prediction task, we employ a predictor equivalent to the decoder, and for the node classification task, we utilize 1 layer of MLP as a classifier.

## 5 EXPERIMENTS

We demonstrate the effectiveness of our proposed model against various baselines on node classification benchmarks and link prediction benchmarks. Then we analyze the contribution of the structural process and feature process of our model.

### 5.1 EXPERIMENTAL SETUP

**Datasets.** To validate our models, we utilize Open Graph Benchmark (OGB) dataset for link prediction tasks and node classification tasks (Hu et al., 2020). We use four OGB link property datasets for link prediction tasks: OGB-PPA, OGB-Collab, OGB-DDI, and OGB-Citation2. OGB-PPA is an undirected and unweighted graph representing protein association. Nodes are proteins from different specifies and edges mean biological associations. Each node feature is a one-hot vector indicating the species to which the protein belongs. OGB-Collab is an undirected graph, which represents a collaboration network where edges denote collaborations between authors. OGB-DDI is an undirected, unweighted graph that contains drug-drug interactions, with edges indicating interactions such as combined effects. Please note that this dataset lacks node features. OGB-Citation2 is a citation network graph with direction. Each node in the graph corresponds to a paper, and a directed edge indicates that one paper cites another. Both OGB-Citation2 and OGB-Collab include node features obtained from embedding models. For node classification tasks, we use three benchmark datasets: OGB-Arxiv, OGB-Products, and PubMed.

**Evaluation.** We evaluate our model with Hits@K metric and Mean reciprocal rank (MRR) in link prediction. Hits@K is based on ranking positive test edges against randomly sampled negative edges. The ranking performance is measured by the ratio of positive test edges ranked at or above the K-th position. In OGB-PPA, the K-th position is set to 100, while for OGB-Collab and OGB-DDI,

Table 1: Link prediction performances on Open Graph Benchmark (OGB) datasets. OOM denotes 'out of memory'. **Bold underline** indicates the best performance and **bold** indicates the second best performance.

| Model | OGB-PPA | OGB-Collab | OGB-DDI | OGB-Citation2 |
|---|---|---|---|---|
| Common Neighbors | $27.65 \pm 0.00$ | $50.06 \pm 0.00$ | $17.73 \pm 0.00$ | $76.20 \pm 0.0$ |
| Adamic Adar | $32.45 \pm 0.00$ | $53.00 \pm 0.00$ | $18.61 \pm 0.00$ | $76.12 \pm 0.0$ |
| Resource Allocation | $\underline{\mathbf{49.33}} \pm 0.00$ | $52.89 \pm 0.00$ | $6.23 \pm 0.00$ | $76.20 \pm 0.0$ |
| Matrix Factorization | $23.78 \pm 1.82$ | $34.87 \pm 0.23$ | $13.29 \pm 2.32$ | $50.48 \pm 3.09$ |
| MLP | $0.99 \pm 0.15$ | $16.05 \pm 0.48$ | N/A | $25.13 \pm 0.28$ |
| GCN | $15.37 \pm 1.25$ | $44.57 \pm 0.64$ | $40.87 \pm 6.08$ | $82.57 \pm 0.26$ |
| GAT | OOM | $41.73 \pm 1.01$ | $32.57 \pm 3.48$ | OOM |
| SAGE | $12.51 \pm 2.02$ | $47.86 \pm 0.64$ | $47.06 \pm 5.21$ | $80.18 \pm 0.15$ |
| JKNet | $11.73 \pm 1.98$ | $47.52 \pm 0.73$ | $\mathbf{57.95} \pm 7.69$ | OOM |
| SEAL | $47.18 \pm 3.60$ | $\underline{\mathbf{54.27}} \pm 0.46$ | $29.86 \pm 4.37$ | $\underline{\mathbf{86.77}} \pm 0.31$ |
| GDM(ours) | $\mathbf{48.32} \pm 0.68$ | $\mathbf{53.82} \pm 0.35$ | $\underline{\mathbf{60.56}} \pm 2.32$ | $\mathbf{84.52} \pm 0.42$ |

it is set to 50 and 20, respectively. The evaluation metric for OGB-Citation2 is MRR. It calculates the reciprocal rank of the true edges within the pool of negative candidates for each source node and then averages these values across all source nodes. To further demonstrate the ability to learn compendious underlying structures in node classification, we constrain a semi-supervised setting by vastly reducing the number of nodes per label in train sets. Under this setting, accuracy measures the performance on OGB-Arxiv, OGB-Products, and PubMed.

**Baselines.** For baselines on link prediction, we include prevalent GNN-based models: GCN (Kipf & Welling, 2017), GAT (Veličković et al., 2018), GraphSAGE (Hamilton et al., 2017), JKNet (Xu et al., 2018), Variational Graph Autoencoder (Kipf & Welling, 2016) and SEAL (Zhang & Chen, 2018). Note that SEAL extracts enclosing subgraph to utilize in link prediction. Additionally, three link prediction heuristics (Liben-Nowell & Kleinberg, 2003; Adamic & Adar, 2003; Zhou et al., 2009), Matrix factorization (Koren et al., 2009), and Multi-layer perceptron (Haykin, 1994) are included in baselines. Baseline models for semi-supervised node classification include GCN, GAT, APPNP (Klicpera et al., 2019), GCNII (Ming Chen et al., 2020), and C&S (Huang et al., 2020).

**Implementation Details.** We implemented link prediction heuristics, such as Common Neighbor(CN), Adamic Adar(AA), and Resource Allocation(RA), based on the paper (Liben-Nowell & Kleinberg, 2003; Adamic & Adar, 2003; Zhou et al., 2009). For GCN, GraphSAGE, GAT, JKNet, APPNP, GCNII, and MLP we used the implementation in PyTorch Geometric (Fey & Lenssen, 2019), and for SEAL and C&S, we used the implementation from the official repository. We trained Graph dissipation model with a 2-layer GDM encoder for OGB-Collab, OGB-DDI, OGB-Arxiv, OGB-Products, and PubMed. Due to memory issues, we trained OGB-PPA, OGB-Citation2 with a 3-layer GDM encoder. Note that we compute normalized Laplacian for numerical stability and we use random sample from dropped edges in denoising task for efficiency. Also, we set diffusion state to 6 for OGB-Collab, OGB-DDI, 10 for OGB-PPA, 3 for OGB-Citation2. For fair comparison, we reported performances of all baselines and GDM as the mean and the standard deviation obtained from 10 independent runs with fixed random seed {0 9}. To simulate more real world-like scenario, we did not use validation edges as input in OGB-Collab. The experiments are conducted on A100(40GB) and A40(48GB).

## 5.2 LINK PREDICTION RESULTS

Table 1 reports the results of OGB link prediction benchmarks. In terms of performance, our Graph dissipation model generally shows improved performance than other baselines. This indicates our GDM is capable of learning latent distribution of underlying factors. Specifically, Graph dissipation model shows the second-best performance which is fairly close to the best performance in OGB-Collab and OGB-PPA, following SEAL and Adamic Adar heuristic, which means OGB-Collab and OGB-PPA have important but hidden structural properties. This implies our GDM captures latent structural factors as well as structure heuristics and SEAL, which is designed to generalize higher-order heuristics. On the other hand, OGB-Citation2 seems to have a latent distribution containing both informative features and structure factors. Our model also showed outperforms the baselines except for SEAL. Note that our GDM still showed the second-best performance without using the

Table 2: Node classification performance on OGB-Arxiv, OGB-Products, and PubMed dataset. OOM denotes 'out of memory'. **Bold** indicates the best performance.

| Model | OGB-Arxiv | | | OGB-Products | | | PubMed | | |
|---|---|---|---|---|---|---|---|---|---|
| Fixed $k$ nodes | $k=1$ | $k=5$ | $k=10$ | $k=1$ | $k=5$ | $k=10$ | $k=1$ | $k=5$ | $k=10$ |
| GCN | $31.69 \pm 2.74$ | $52.97 \pm 0.94$ | $58.39 \pm 0.50$ | $38.93 \pm 2.09$ | $62.69 \pm 1.27$ | $66.23 \pm 0.91$ | $45.87 \pm 2.44$ | $60.56 \pm 1.44$ | $69.50 \pm 0.68$ |
| GAT | $25.60 \pm 2.95$ | $50.87 \pm 1.78$ | $57.23 \pm 0.75$ | $35.81 \pm 2.42$ | $60.72 \pm 1.93$ | $64.80 \pm 1.21$ | $43.57 \pm 2.71$ | $58.38 \pm 2.06$ | $68.40 \pm 1.49$ |
| APPNP | $29.36 \pm 2.19$ | $52.47 \pm 1.26$ | $56.42 \pm 0.83$ | $36.35 \pm 2.20$ | $63.01 \pm 2.10$ | $66.85 \pm 0.84$ | $43.04 \pm 1.72$ | $56.94 \pm 1.90$ | $69.99 \pm 0.73$ |
| GCNII | $30.94 \pm 2.30$ | $51.94 \pm 1.18$ | $57.65 \pm 0.94$ | $33.64 \pm 2.32$ | $61.43 \pm 2.36$ | $64.90 \pm 1.39$ | $43.29 \pm 2.53$ | $56.18 \pm 1.84$ | $70.60 \pm 0.93$ |
| C&S | $30.63 \pm 1.88$ | $51.73 \pm 1.30$ | $56.57 \pm 1.43$ | $40.47 \pm 1.97$ | $62.18 \pm 1.57$ | $67.53 \pm 1.40$ | $44.91 \pm 1.24$ | $57.44 \pm 1.36$ | $68.78 \pm 1.07$ |
| GDM(ours) | $\mathbf{38.40 \pm 1.64}$ | $\mathbf{57.22 \pm 0.85}$ | $\mathbf{60.97 \pm 0.40}$ | $\mathbf{48.96 \pm 1.81}$ | $\mathbf{67.03 \pm 1.05}$ | $\mathbf{70.22 \pm 0.69}$ | $\mathbf{53.06 \pm 1.53}$ | $\mathbf{66.79 \pm 0.92}$ | $\mathbf{72.42 \pm 0.71}$ |

Table 3: Ablation study analyzing the efficacy of each component of the coupled diffusion process.

| Dataset | GDM (original) | GDM w/o feature process | GDM w/o structure process |
|---|---|---|---|
| OGB-Collab | $53.86 \pm 0.35$ | $46.31 \pm 2.35$ | $44.43 \pm 2.91$ |
| OGB-PPA | $49.32 \pm 0.68$ | $25.15 \pm 4.12$ | $20.24 \pm 3.56$ |

full graph to train GDM. GDM achieves the best performance on OGB-DDI, where SEAL shows poor performance. This can be interpreted as SEAL is more focused on capturing structural information while OGB-DDI requires feature learning to investigate important latent factors. Since our model shows improved performance whether the dataset is more dependent on feature or structure, this implies our GDM reasonably captures the integrated and comprehensive latent distribution of a graph.

## 5.3 SEMI-SUPERVISED NODE CLASSIFICATION RESULTS

We conduct experiments on semi-supervised node classification benchmark datasets to validate the effectiveness of GDM on learning node embeddings. We constrained the training index by the fixed $k$ nodes per label. The number $k$ is set to $1, 5, 10$. Table 2 shows the performance of a semi-supervised node classification task that is extremely limited to label scarcity. GDM outperforms other baselines on all datasets and settings. C&S is known to show high accuracy in node classification tasks due to its correlation propagation scheme, however, it seems fairly low performance in this setting. One possible implication is that C&S employs label propagation which may require a minimum number of nodes. According to the results, GDM is effectively captures the latent distribution of nodes, even under very constrained conditions.

## 5.4 ABLATION STUDY

We empirically validate the efficacy of each component in Graph dissipation model through ablation experiments. First, we evaluate GDM without the feature diffusion process and structural diffusion process and evaluate the average performance on link prediction tasks. OGB-Collab requires models to learn both feature and structural hidden representation from a graph. GDM without feature diffusion process and GDM without structural diffusion process both shows degraded performance on OGB-Collab. Similarly, in OGB-PPA, which seems to have important structural latent factors, GDM without structural process shows a slightly larger degradation in the performance. It is interesting that the gap between GDM without feature process and GDM without structure process is larger in OGB-PPA.

## 6 CONCLUSION

In this paper, we introduced the Graph dissipation model (GDM) as a novel approach to learn latent factors of graph-structured data, regarding specifics of various network graph learning tasks. GDM defines Laplacian smoothing as noise during the forward process and lifts dissipation to a structure to capture latent factors that are comprehensive to network graph learning tasks. In future work, we plan to further develop GDM by focusing on learning interpretable latent distribution.

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

# A  APPENDIX

## A.1  DERIVATION OF LOSS FUNCTION OF GDM

This section provides a derivation of the variational lower bound (ELBO) and the loss function of Graph dissipation model (GDM).

Let $G_0$ be a given observed graph data consisting of (X, A), denoting node features and adjacency matrix, respectively. Taking negative log-likelihood, we get

$$\log p_\theta(G_0) \leq -\int q(G_{1:T}|G_0) \log p_\theta(G_0|G_{1:T}) \frac{p_\theta(G_{1:T})}{q(G_{1:T}|G_{G_0})} dG \tag{13}$$

$$= \mathbb{E}_{q(G_{1:T}|G_0)}\left[-\log \frac{p_\theta(G_{0:T})}{q(G_{1:T}|G_0)}\right]. \tag{14}$$

The variational lower bound (ELBO) is obtained as follows:

$$\mathbb{E}_{q(G_{1:T}|G_0)}\left[-\log \frac{p_\theta(G_{0:T})}{q(G_{1:T}|G_0)}\right] = \mathbb{E}_{q(G_{1:T}|G_0)}\left[-\log \frac{p_\theta(G_T)\prod_{t=1}^{T} p_\theta(G_{t-1}|G_t)}{\prod_{t=1}^{T} q(G_t|G_0)}\right] \tag{15}$$

$$= \mathbb{E}_{q(G_{1:T}|G_0)}\left[-\log \frac{p_\theta(G_T)}{q(G_T|G_0)} - \log \prod_{t=2}^{T} \frac{p_\theta(G_{t-1}|G_t)}{q(G_{t-1}|G_0)} - \log p_\theta(G_0|G_1)\right] \tag{16}$$

$$= \mathbb{E}_{q(G_{1:T}|G_0)}\left[-\log \frac{p(G_T)}{q(G_T|G_0)} - \sum_{t=2}^{T} \log \frac{p_\theta(G_{t-1}|G_t)}{q(G_{t-1}|G_0)} - \log p_\theta(G_0|G_1)\right] \tag{17}$$

$$= \sum_{t=2}^{T} \mathbb{E}_q D\left[q(G_{t-1}|G_0)\|p_\theta(G_{t-1}|G_t)\right] + \mathbb{E}_q\left[-\log p_\theta(G_0|G_1)\right] \tag{18}$$

The first term is not trainable as it equals constant and the second term is KL divergence $D_{KL}[q(G_{t-1}|G_0)\|P_\theta(G_{t-1}|G_t)]$. Posterior $q(G_{t-1}|G_0)$ cannot be expressed in a closed-form solution because $q(G_{t-1}|G_0)$ is unknown as we did not define a prior distribution on a network graph. Several well-known distributions are inadequate to define a network graph due to its structural characteristics. Thus, posterior $q(G_{t-1}|G_0)$ is intractable.

However, we can approximate $q(G_{t-1}|G_0)$ by decomposing $G$ into features $X$ and a structure $A$. We can rewrite the second term into

$$D\left[q(G_{t-1}|G_0)\|p_\theta(G_{t-1}|G_t)\right] = D\left[q(X_{t-1}|X_0)\|p_\theta(X_{t-1}|X_t)\right] + D\left[q(A_{t-1}|A_0)\|p_\theta(A_{t-1}|A_t)\right]. \tag{19}$$

Recall the forward process of GDM,

$$q(X_{1:T}|X_0) = \prod_{t=1}^{T} q(X_t|X_0), \quad q(X_t|X_0) := (\mathbf{I} - \boldsymbol{L})^t \boldsymbol{X}_0 \tag{20}$$

$$q(A_{1:T}|A_0) = \prod_{t=1}^{T} q(A_t|A_{t-1}), \quad q(A_t|A_{t-1}) := \mathcal{B}(\boldsymbol{A}_t|\boldsymbol{A}_{t-1}, s(\hat{\boldsymbol{X}}_t)). \tag{21}$$

Based on Eq.20, $D\left[q(X_{t-1}|X_0)\|p_\theta(X_{t-1}|X_t)\right]$ indicates deblurring smoothed features,

$$D\left[q(X_{t-1}|X_0)\|p_\theta(X_{t-1}|X_t)\right] = \|f_\theta(X_t, A_t) - X_{t-1}\|_2^2.$$

which is equivalent to predicting less smoothed features. Note that we lift feature dissipation to the forward structural process, under the mild assumption, approximated $q(A_t|A_{t-1}) \approx q(A_t|A_0)$ can be used as follows:

$$q(A_{ij}^{(t-1)}|A_{ij}^{(0)}) = \begin{cases} \mathcal{B}(A_{ij}^{(t-1)}; p \overset{\propto}{\sim} LX = I - (I - LX)), & \text{if } A_{ij}^{(0)} = 1 \\ \mathcal{B}(A_{ij}^{(t-1)}; p = 0), & \text{if } A_{ij}^{(0)} = 0 \end{cases} \tag{22}$$

Only in the first case, edge existence probability $p$ is uncertain. Note that edge probability $p$ is correlated to Laplacian matrix which feature dissipation relied on. To make the graph structure sparser as the node features converge to oversmoothing, we defined the forward structural process with stochastic structure sampling dependent on features. However, the edge probability $p$ estimation has uncertainty because we lift the feature distance upon edge existence probability through the forward structural process. The intuition behind the forward structural process is to lift signal dissipation to a graph structure. Leveraging this intuition, the edge probability $p$ can be estimated by discrepancy of structural information which implies dissipation on a graph structure. Therefore, $D\left[q(A_{t-1}|A_0)\|p_\theta(A_{t-1}|A_t)\right]$ is approximated with a discrepancy between $L$ and $L_{t-1}$,

$$D\left[q(A_{t-1}|A_0)\|p_\theta(A_{t-1}|A_t)\right] = \|f_\theta(X_t, A_t) - (L_0 - L_{t-1})\|_2^2$$

predicting the discrepancy between graph Laplacian where dissipation is dependent.

Finally, we obtain the loss function as follows:

$$-\log p(G_0) \leq \mathbb{E}_{q(G_{1:T}|G_0)}\left[-\log\frac{p_\theta(G_{0:T})}{q(G_{1:T}|G_0)}\right] \tag{23}$$

$$= \mathbb{E}_{q(G_{1:T}|G_0)}\left[-\log\frac{\cancel{p(G_T)}}{\cancel{q(G_T|G_0)}} - \sum_{t=2}^{T}\log\frac{p_\theta(G_{t-1}|G_t)}{q(G_{t-1}|G_0)} - \log p_\theta(G_0|G_1)\right] \tag{24}$$

$$= \sum_{t=2}^{T}\mathbb{E}_q D\left[q(X_{t-1}|X_0)\|p_\theta(X_{t-1}|X_t)\right] + \sum_{t=2}^{T}\mathbb{E}_q D\left[q(A_{t-1}|A_0)\|p_\theta(A_{t-1}|A_t)\right]$$
$$+ \mathbb{E}_q\left[-\log p_\theta(X_0|X_1)\right] + \mathbb{E}_q\left[-\log p_\theta(A_0|A_1)\right] =: \mathcal{L}_{\text{GDM}} \tag{25}$$

Based on the aforementioned approximation of $q(X_{t-1}|X_0)$ and $q(A_{t-1}|A_0)$, the loss function becomes

$$\mathcal{L}_{\text{GDM}} = \beta_t \sum_{t=2}^{T}\|f_\theta(X_t, A_t) - X_{t-1}\|_2^2 + \gamma\sum_{t=2}^{T}\|f_\theta(X_t, A_t) - (L_0 - L_{t-1})\|_2^2$$
$$+ \beta_1\|f_\theta(X_1, A_1) - X_0\|_2^2 + \lambda\text{BCE}(f_\theta(X_1, A_1), A_0). \tag{26}$$

$\beta_t, \beta1, \gamma$ and $\lambda$ are weighting hyperparameters.

## A.2 HYPERPARAMETER SENSITIVITY ANALYSIS

We analyze Graph dissipation model (GDM) to demonstrate how hyperparameters affect the performance of GDM. We conduct experiments with 4 hyperparameters in GDM loss function, $\mathcal{L}_{\text{GDM}}$. $\beta_t, \beta_1, \gamma$ and $\lambda$ is weighting hyperparameters for $\mathcal{L}_{\text{feat}}, \mathcal{L}_{\text{feat-recon}}, \mathcal{L}_{\text{Lap}}$, and $\mathcal{L}_{\text{recon}}$, respectively. We measure Hits@50 by changing one hyperparameter while the rest of hyperparameters are fixed to the best value. The result (Figure. 2) demonstrates that GDM is fairly robust to hyperparameters that weight the components of GDM loss $\mathcal{L}_{\text{GDM}}$.

## Hyperparameter Senstivity Analysis

Figure 2: Visualization of hyperparameter sensitivity analysis on OGB-Collab.

## B  PRELIMINARY OF OVER-SMOOTHING

Laplacian smoothing is written in the matrix formulation as

$$X' = (I - \lambda D^{-\frac{1}{2}} L D^{-\frac{1}{2}})X = (I - \lambda L_{sym})X,$$
$$X' = (I - \lambda D^{-1} L)X = (I - \lambda L_{RW})X,$$

where $I$ denotes the identity matrix. Laplacian smoothing produces the diffusion of signal across the graph, leading to a filtered representation of the signal on the graph structure with respect to neighborhood nodes' features. Note that Laplacian smoothing can be applied iteratively to propagate the signal on the graph further, gradually blurring node representations.

**Corollary B.1** *(Li et al., 2018) For any $\boldsymbol{x} \in \mathbb{R}^d$ and $0 < \lambda \leq 1$, a graph without bipartite components converges to a linear combination of $\{\boldsymbol{1}^{(c)}\}_{c=1}^C$:*

$$\lim_{l \to \infty} (1 - \lambda L_{sym})^l \boldsymbol{x} = D^{-\frac{1}{2}} [\|_{c=1}^C \boldsymbol{1}^{(c)}] \boldsymbol{w},$$
$$\lim_{l \to \infty} (1 - \lambda L_{RW})^l \boldsymbol{x} = [\|_{c=1}^C \boldsymbol{1}^{(c)}] \boldsymbol{w},$$

*where $\boldsymbol{w} \in \mathbb{R}^c$, and $c$ indicates connected components.*

