# OpenReview forum: "Denoising Graph Dissipation Model Improves Graph Representation Learning"
_ICLR.cc/2024/Conference — Submitted to ICLR 2024_

### Official Review · Reviewer_yTz6 · 2023-10-31

**Soundness:** 2 fair
**Presentation:** 3 good
**Contribution:** 2 fair
**Rating:** 6
**Confidence:** 4

**Summary:**

Existing graph representation learning methods mainly focus on task-specific factors rather than universal factors that can be used for any downstream tasks. This work proposes Graph Dissipation Model (GDM) to learn the latent intrinsic distributions of the graph based on the diffusion models, which enables the learned representations to be utilized for any downstream tasks. To encode both node feature and structural information, GDM introduces a coupled diffusion model framework consisting of a feature diffusion process and a structure diffusion process. Laplacian smoothing is innovatively used as a noise source for the feature diffusion process and edge removal is also defined as a noise source for the structure diffusion process. Experiments on both link prediction and node classification show that GDM achieves comparable performance for existing graph representation learning baselines on both tasks, demonstrating GDM's capability of learning universal factors that can be applied to any downstream tasks.

**Strengths:**

1. This work proposes GDM, the first diffusion-based graph representation learning model that encodes both node feature and structure information. GDM is able to learn comprehensive and universal latent structures from a graph without explicit bias for specific tasks.

2. The idea of utilizing Laplacian smoothing as a noise source for the feature diffusion process and over-smoothing as a convergence state is novel and interesting. Such a design for blurring node features is also more natural in the graph learning setting.

3. Experiments indicate that GDM achieves comparable performance on the link prediction task compared to baselines, and outperforms baselines on a semi-supervised node classification with few training labels, demonstrating that GDM learns universal graph representations that can be applied to downstream tasks.

**Weaknesses:**

1. Although GDM aims to learn comprehensive and universal graph representations, Equation 10 in the paper still contains the downstream task loss as a part of the final loss. I wonder if GDM without downstream task loss can learn universal graph representations, or we should regard GDM as a universal framework that can incorporate any downstream task loss. Have the authors done some experiments to evaluate the universal graph representations obtained by GDM without downstream task loss?

2. In this work, the authors did not mention the time complexity of GDM and its runtime in experiments. As GDM requires eigendecomposition of the graph Laplacian matrix, I wonder if the authors could further discuss GDM's time complexity and also provide some results of the GDM's runtime compared to other baselines in the link prediction and node classification experiments.

3. (Minor) I did not find any supplementary materials discussing the details of the implementation of GDM and the experiments conducted in the paper. There is also no code implementation of GDM to reproduce the experimental results presented in the paper.

4. (Minor) Typo: In the Implementation Details of Section 5.1, \
"Also we set iffusion state to 3 for OGB-Citation2" $\rightarrow$ "Also we set diffusion state to 3 for OGB-Citation2"

**Questions:**

1. Please see the questions mentioned in the Weaknesses.

2. As the over-smoothing issue appears after only several Laplacian smoothing operations (i.e., node representations converge to identical after only several steps), it seems the value of time step $t$ can be small if we set the over-smoothing as the convergence state. Therefore, I wonder how to choose a proper $t$ to ensure sufficient diffusion and if the authors have done some experiments on the selection of $t$.

---

> ### Author Response · Authors · 2023-11-21
> **Response to Reviewer yTz6 - Weakness (1), (2), Question (2), Minor**
>
> Thank you very much for dedicating time to review our submission and providing crucial comments! We are pleased to address your concerns as follows:
>
> > Although GDM aims to learn comprehensive and universal graph representations, Equation 10 in the paper still contains the downstream task loss as a part of the final loss. I wonder if GDM without downstream task loss can learn universal graph representations, or we should regard GDM as a universal framework that can incorporate any downstream task loss. Have the authors done some experiments to evaluate the universal graph representations obtained by GDM without downstream task loss?
>
> This is a great point!
> GDM is "a universal framework that can incorporate any network graph downstream task loss", which accurately expresses our approach.
> The motivation behind GDM is that existing graph representation learning models are investigated for improving particular graph representation learning tasks, yet they are often vulnerable to other tasks. For instance, in the link prediction task, some existing models assume generalizing some graph heuristics to find missing links. However, those existing models are not capable of learning node embeddings for node classification tasks.
>
> Consequently, when we refer to “comprehensive and universal graph representations”, which GDM aims to learn, it intends to convey a representation that is universally applicable to various network graph representation learning downstream tasks while naturally regarding specifics of those tasks without injecting task-dependent assumptions or biases. Note that it covers both features and structure information that are entailed from a given task. Unlike GDM, existing graph representation learning methods have focused on defining task-dependent / task-specific assumptions for particular graph learning tasks. For instance, in the link prediction task, some existing models assume generalizing some graph heuristics to find missing links that are not inferred by node embeddings. In node classification tasks, some approaches introduced biases that enhanced intraclass relationships and weakened interclass relations to improve node classification performance. Thus, there are no powerful models for both link prediction and node classification tasks. However, our GDM captures comprehensive latent factors needed for a given task faithfully without relying on such task-specific assumptions, achieving competitive performance in both tasks.
>
> To the best of our knowledge, we initially raised such motivation and research for diffusion model for graph representation learning regarding both graph feature and structure has not been studied before, highlighting the novelty of our study.
>
> > In this work, the authors did not mention the time complexity of GDM and its runtime in experiments. As GDM requires eigendecomposition of the graph Laplacian matrix, I wonder if the authors could further discuss GDM's time complexity and also provide some results of the GDM's runtime compared to other baselines in the link prediction and node classification experiments.
>
> Since we leverage Laplacian smoothing as a noise source in the forward process of GDM, it does not require the eigendecomposition of graph Laplacian matrix. We mentioned eigendecomposition for providing insight on graph spectral domain. We are sorry for the misleading typo. We will immediately revise it.
> As there is no algorithm in the training procedure, GDM's runtime is 55seconds on OGB-Collab which is comparable with conventional GNNs which take 20~30 seconds on the dataset. It is notable that GDM achieves competitive results in link prediction tasks compared to SEAL or Neo-GNNs where runtime is fairly longer ($>3 \text{min}$) due to extracting enclosing subgraphs or generalizing algorithms.
>
> >  As the over-smoothing issue appears after only several Laplacian smoothing operations (i.e., node representations converge to identical after only several steps), it seems the value of time step $t$
>  can be small if we set the over-smoothing as the convergence state. Therefore, I wonder how to choose a proper
>  to ensure sufficient diffusion and if the authors have done some experiments on the selection of $t$.
>
> This is a great point! Right. Unlike diffusion models proposed for image or graph generation, GDM does not require extremely large numbers as time step values, such as 1000 or 10000. The oversmoothing issue in GNNs is known to occur when there are at least five layers. Based on this, we determined that setting the time step ($T$) to a minimum of 5 is sufficient. Through experiments, we found that setting $T$ to 6 for Collab, PPA, and DDI is sufficient. Adaptive learning of $T$ value is something we consider for future work.
> Additionally, we will include the sensitivity analysis for the selection of $T$ in the appendix.
>
> *Minor*
> (3) We will upload the code implementation of GDM soon.
> (4) Thanks to the reviewer, we will fix typos in the revised paper.

---

> > ### Comment · Reviewer_yTz6 · 2023-11-23
> > **Official Comment by Reviewer yTz6**
> >
> > Thank you the authors for the response. Regarding the question "a universal framework that can incorporate any network graph downstream task loss", what I was trying to ask is whether the authors have done some experiments on some experiments to evaluate the universal graph representations obtained by GDM without downstream task loss, but the authors did not answer my question directly. However, I understand the general motivation of GDM and I would like to keep my score as is.

---

> ### Author Response · Authors · 2023-11-23
>
> We would like to answer the reviewer's question about experiments to evaluate the universal graph representation.
> We did not conduct experiments on evaluating the universal graph representation since what we mean "universal" is that GDM is universally applicable to network graph representation learning tasks and shows competitive performance on both major graph representation learning tasks.
>
> Also, we uploaded the revised paper.
> We revised some points the reviewer suggested.
> - More detailed explanation of using Laplacian smoothing as noise source to diffuse and dissipate graph signal (Section 4)
> - Revise some confusing notations, terms, equations into more precise and consistent expressions (Section 1, 4)
> - Clarify motivation and novelty of GDM (Section 1)
> - Clarify GDM is for network graph representation learning (Section1)
> - Derivation of loss function (Appendix)
> - Hyperparameter sensitivity analysis (Appendix)
>
> We hope that our response addresses the reviewer's concern and leads to stronger support.
> We sincerely appreciate the feedback to improve the paper.

---

### Official Review · Reviewer_ntqg · 2023-11-01

**Soundness:** 3 good
**Presentation:** 3 good
**Contribution:** 3 good
**Rating:** 3
**Confidence:** 4

**Summary:**

This paper introduces the Graph Dissipation model which is a coupled diffusion model operating on node feature and graph structure space simultaneously. The model utilizes the Laplacian smoothing to get the noised node features, promoting the denoising network to capture the structural information during training. The evaluation tasks include link prediction and node classification.

**Strengths:**

- The paper is well-written and easy to follow.
- Using Laplacian smoothing to diffuse the node features is an interesting operation which sounds technique.
- Experiments support the statements in the paper.

**Weaknesses:**

- The novelty of the structure diffusion process with randomly removing edges is limited, which also appears in [3]. Further, this reverse process of the used structure diffusion cannot correspond to the forward process.
- In Eq(9), the Feature prediction loss and structure dissipation loss are both confusing. How to calculate the $q(X_{t-1}|X_{t},X_{0})$ and $q(A_{t-1}|A_{t},A_{0})$？ The relationship between ELOB(Eq. (8)) and final loss (Eq 9) should rigorously prove.
- Eq (6) is confusing since the $A_t$ is sampled from eq 5, which is unrelated to $A_{t-1}$. So. How to calculate the elements of $A_t$?
- The experimental results show the proposed method doesn’t achieve competitive performance in Link prediction (https://ogb.stanford.edu/docs/leader_linkprop/). Some important baselines are missing, such as GIN, on the node classification task.

Minor concerns:
- Eq (9) is out of bounds.
- The claim “there has been no work on diffusion models for graph representation learning in both feature and structural aspects” is inappropriate because there exist related works such as MoleculeSDE[1],[2].
- The formula at the bottom of page 3 lacks of the explanation of $x$.
- Eq. (8) should be an inequality.
- Is there  $\zeta $ in Eq(5)?

[1] A Group Symmetric Stochastic Differential Equation Model for Molecule Multi-modal Pretraining.

[2] Fast Graph Generation via Spectral Diffusion

[3] Efficient and Degree-Guided Graph Generation via Discrete Diffusion Modeling

**Questions:**

- From the Leaderboards of OGB(https://ogb.stanford.edu/docs/leader_linkprop/), the experimental results of this paper are not very competitive. Why the GDM don’t use a powerful GNN as the denoising network? In my understanding, the Loss $L_{diff}$ can be used in any GNN for graph representation learning.
- What is the relationship between GDM and Digress[1]? The GDM seems to be a specific case of Digress.
- What is the benefit of samping $A_{t}$ from Eq(5) instead of a random transition from $A_{t-1}$ like [2]

[1] DIGRESS: DISCRETE DENOISING DIFFUSION FOR GRAPH GENERATION

[2] Diffusion Models for Graphs Benefit From Discrete State Spaces

---

> ### Author Response · Authors · 2023-11-21
> **Response to Reviewer ntqg - Weakness(1), (3), Question(3)**
>
> Thank you very much for dedicating time to review our submission and providing thoughtful comments! We are pleased to address your concerns as follows:
>
> > The novelty of the structure diffusion process with randomly removing edges is limited, which also appears in [3]. Further, this reverse process of the used structure diffusion cannot correspond to the forward process.
>
> We acknowledge that the term 'edge removal' can cause misleading and it does not express the goal of the process.
>
> Our edge removal is completely different from random edge removal from [3].
> Because we aim to reflect the decay of signal/information on the graph, we designed the subgraph sampling that can lift feature decay to the graph structure.
> Specifically, edges are sampled to be dropped from the given graph structure stochastically based on the amount of feature smoothed as drop parameter $p$ during the structure diffusion process.
> This is significantly different from [3] as this sampling is coupled with the feature diffusion process and incorporates removing edges in a way that is influenced by the degree of feature smoothing. This sampling procedure is proposed to lift dissipation of features to the structure.
>
> On the contrary, [3] employs edge removal for generic graph generation by randomly selecting nodes and then probabilistically removing edges connected to those selected nodes solely relying on randomness. Edge removal of [3] may be suitable for graph generation since it requires diverse structures to generate unique structures but with similar graph statistics.
>
> To clarify the novelty of GDM and to prevent misleading, we will use the term "dissipative structure sampling" instead of the term "edge removal" in the revised paper.
>
> > Eq (6) is confusing since the $A_t$ is sampled from eq 5, which is unrelated to $A_{t-1}$. So. How to calculate the elements of $A_t$?
>
> Sorry for the confusion.
> Eq.(5) expresses each edge is sampled to be dropped by the parameter $p$, lifting dissipation of feature to the graph structure. We meant to show Bernoulli sampling on each edge.
> Eq.(6) expresses the sampling for whole adjacency $A_t$.
>
> We will clarify the Eq.(5) as follows:
>
> $ A\_{t}[ij] \sim \text{Bern}(A_{t} | A_{t-1}[ij]=1, p=s(  \hat{\mathbf{x}}_{i}^{(t-1)} , \hat{\mathbf{x}}\_{j}^{(t-1)} )). $
>
> > What is the benefit of samping $A_{t}$ from Eq(5) instead of a random transition from $A_{t-1}$ like [2]?
>
> Since a network graph is complicated to be defined with some family of known distribution, we designed GDM to learn underlying factor distribution by the concept of ‘dissipation’. In the forward process, GDM uses Laplacian smoothing as a noise source, gradually converging into oversmoothing. We consider the decrease in the differences between node features as the dissipation of signal (in spectral) or feature information (in spatial). To learn feature-structure integrated representations, we lift the dissipating of features to the graph structure. This is why we define structure sampling as Eq. (5) and Eq. (6).
> On the contrary, [2] appears to employ random flipping to learn diverse structures as many as possible, as the model targets only generic graph generation. The motivation that GDM addresses is different from [2], thus, we defined our own sampling method, ‘Dissipative structure sampling’ for GDM that aligns with our motivation.

---

> ### Author Response · Authors · 2023-11-22
> **Response to Reviewer ntqg - Weakness (2)-1**
>
> > In Eq(9), the Feature prediction loss and structure dissipation loss are both confusing. How to calculate the $ q( X\_{t-1} | X_{t}, X\_{0} ) $ and $ q( A\_{t-1} | A_{t}, A\_{0} ) $ ? The relationship between ELOB(Eq. (8)) and final loss (Eq 9) should rigorously prove.
>
> This is a good point. First, we will derive the loss (Eq. 9) from the negative of evidence lower bound (ELBO), then explain q( X\_{t-1} | X_{t}, X\_{0} ) $ and $ q( A\_{t-1} | A_{t}, A\_{0} ) $.
>
> Let $G\_0$ be a given observed graph data consisting of $(X, A), denoting node features and adjacency matrix, respectively. Taking negative log-likelihood with evidence lower bound (ELBO), we get
>
> $$
>  -\log p\_{\theta}(G_{0}) \leq  -\int q(G_{1:T} | G_{0}) \log p_{\theta}(G_{0}|G_{1:T}) \frac{p_{\theta}(G_{1:T})}{q(G_{1:T}|G_{G\_{0}})} \\,dG
> $$
> $$
> = \mathbb{E}\_{q(G_{1:T} | G_{0})} \left[ -\log \frac{p_{\theta}(G_{0:T})}{q(G_{1:T}|G\_{0})} \right] .
> $$
>
> Note that the forward feature process is defined as $ q(X\_{1:T} | X_{0}) = \prod_{t=1}^{T} q(X_{t} | X\_{0}) $. As we lift feature dissipation to the forward structural process, under mild assumption, approximated $q(A\_{t}|A_{t-1}) \approx q(A_{t}|A\_{0}) $ can be used as follows:
> $$
> = \mathbb{E}\_{q(G_{1:T} | G_{0})} \left[ -\log \frac{p_{\theta}(G_{T}) \prod_{t=1}^{T} p_{\theta}(G_{t-1}|G_{t})}{\prod_{t=1}^{T} q(G_{t}|G\_{0})} \right]
> $$
> $$
> = \mathbb{E}\_{q(G_{1:T} | G_{0})} \left[ -\log \frac{p_{\theta}(G_{T})}{q(G_{T}|G_{0})} -\log \prod_{t=2}^{T} \frac{p_{\theta}(G_{t-1}|G_{t})}{q(G_{t-1}|G_{0}) } -\log p_{\theta}(G_{0}|G\_{1}) \right]
> $$
> $$
> = \mathbb{E}\_{q(G_{1:T} | G_{0})} \left[ -\log \cancel{\frac{p(G_{T})}{q(G_{T}|G_{0})}} -\sum_{t=2}^{T} \log \frac{p_{\theta}( G_{t-1}| G_{t} ) }{q(G_{t-1}|G_{0})} -\log p_{\theta}(G_{0} | G\_{1}) \right].
> $$
> The second term can be interpreted as KL divergence $D\_{KL} [ q(G_{t-1}|G_{0}) \Vert P_{\theta}(G_{t-1}|G\_{t}) ]$. Posterior $q( G\_{t-1} | G\_{0})$ cannot be expressed in a closed-form solution because $q(G\_{t-1}|G\_{0})$ is unknown as several well-known distributions are inadequate to define a network graph due to its structural characteristics.
> Especially, it becomes more difficult because GDM targets to learn feature-structure integrated latent representation. Therefore, posterior $q( G\_{t-1} | G\_{0})$ is intaractable.
>
> However, we define $\mathbb{E}\_{q(G_{1:T} | G_{0})} \left[ -\sum_{t=2}^{T} \log \frac{p_{\theta}( G_{t-1}| G_{t} ) }{q(G_{t-1}|G\_{0})} \right]$ by decomposing $G$ to feature $X$ and structure $A$.
> Posterior on feature $q(X\_{t-1}|X\_{0})$ is straight forward according to Eq.(3): $q(X\_{t-1}|X_{0}) = (I - L)^{t-1}X\_{0}$.
> Thus, we define $\mathbb{E}\_{q(X_{1:T} | X_{0})} \left[ -\sum_{t=2}^{T} \log \frac{p_{\theta}( X_{t-1}| X_{t} ) }{q(X_{t-1}|X\_{0})} \right]$ as predicting $(I-L)^{t-1}X_{0}$ from $X\_{t}$, i.e., $\lVert f_{\theta}(X_{t}, A_{t})-X\_{t-1} \rVert\_{2}^{2}$.
>
> Posterior on structure $q(A\_{t-1}|A\_{0})$ is approximately obtained as follows:
> $$
> q(A\_{ij}^{(t-1)}|A_{ij}^{(0)}) =
> \mathcal{B}(A_{ij}^{(t-1)};p \propto  L^{t-1}X), \quad  \text{if } A\_{ij}^{(0)}=1
> $$
> $$
> q(A\_{ij}^{(t-1)}|A_{ij}^{(0)}) =
> \mathcal{B}(A_{ij}^{(t-1)};p=0),  \quad  \text{if }A\_{ij}^{(0)}=0
> $$
> Only in the first case, edge existence probability $p$ is uncertain.
> Note that edge probability $p$ is correlated to Laplacian matrix which feature dissipation relied on.
> To learn comprehensive and integrated latent representation, we defined the forward structural process with stochastic structure sampling based on connected node pairs' similarity. Consequently, the forward structural process makes the graph structure sparser as the node features converge to oversmoothing. However, due to its uncertainty, the edge probability $p$ cannot be estimated by just restoring feature similarity.
> The intuition behind the forward structural process is to lift signal dissipation to graph structure. Leveraging this intuition, the edge probability $p$ is estimated by discrepancy of structural information which implies dissipation upon graph structure. Therefore, we define $ \mathbb{E}\_{q(A_{1:T} | A_{0})} \left[ -\sum_{t=2}^{T} \log \frac{p_{\theta}( A_{t-1}| A_{t} ) }{q(A_{t-1}|A\_{0})} \right]$ with predicting the discrepancy between graph Laplacian where dissipation is dependent, i.e., $\lVert f_{\theta}(X_{t}, A_{t})-(L_{0}-L_{t-1})  \rVert\_{2}^{2}$.

---

> > ### Author Response · Authors · 2023-11-22
> > **Response to Reviewer ntqg - Weakness (2)-2, Minor (1,3,4,5)**
> >
> > Finally, we obtain the loss function as follows:
> > $$
> > -\log p(G\_{0}) \leq \mathbb{E}\_{q(G_{1:T} | G_{0})} \left[ -\log \cancel{\frac{p(G_{T})}{q(G_{T}|G_{0})}} -\sum_{t=2}^{T} \log \frac{p_{\theta}( G_{t-1}| G_{t} ) }{q(G_{t-1}|G_{0})} -\log p_{\theta}(G_{0} | G\_{1}) \right] =: L\_{diss}
> > $$
> >
> > $$
> > L\_{diss} =  \sum\_{t=2}^{T} \mathbb{E}\_{q} D \left[ q(X\_{t-1}| X\_{0}) \Vert p\_{\theta} (X\_{t-1}|X\_{t}) \right]  + \sum\_{t=2}^{T} \mathbb{E}\_{q} D \left[ q(A\_{t-1}| A\_{0}) \Vert p\_{\theta} (A\_{t-1}|A\_{t}) \right]  +  \mathbb{E}\_{q} \left[ -\log p\_{\theta} (X\_{0} | X\_{1}) \right] + \mathbb{E}\_{q} \left[ -\log p\_{\theta} (A\_{0} | A\_{1}) \right]
> > $$
> > Note that $D$ denotes an arbitrary divergence for an unknown latent distribution in this case.
> >
> > To prevent misleading and confusion, we provide a more precise loss formula.
> > Based on the aforementioned approximation of $q(X\_{t-1} | X\_{0})$ and $ q(A\_{t-1} | A\_{0}) $, the loss function becomes
> > $$
> > L\_{diss} =  \beta_{t} \sum\_{t=2}^{T} \Vert f\_{\theta}(X_{t}, A_{t}) - X_{t-1} \Vert_{2}^{2} + \gamma \sum\_{t=2}^{T}  \Vert f\_{\theta}(X_{t}, A_{t}) - (L_{0} - L_{t}) \Vert_{2}^{2} +
> > \beta_{0} \Vert f\_{\theta}(X_{1}, A_{1}) - X_{0} \Vert_{2}^{2} + \lambda \text{BCE}(f_{\theta}(X_{1}, A_{1}), A_{0}).
> > $$
> > $ \beta_{t}, \beta{1}, \gamma, \text{and} \lambda$ are weighting hyperparameters.
> >
> > > Eq (9) is out of bounds. Eq. (8) should be an inequality.
> >
> > To address the reviewer's concern, we provide more precise expressions for the loss function and its derivation. Besides, we acknowledge there are some typos that may cause misleading and confusion. We will correct typos and provide the derivation for loss function in the revised paper including appendix.
> >
> > > The formula at the bottom of page 3 lacks of the explanation of $x$.
> >
> > $x$ located at the bottom of page 3 denotes an arbitrary data point to explain Laplacian smoothing. Since this equation is trivial, we will shorten this part in the revised paper.
> >
> > > Is there $\zeta$ in Eq(5)?
> >
> > $\hat{x}_{i}^{(t-1)}$ denotes a row vector corresponding to node $i$ in $\hat{X}_{t-1}$ in Eq.(4). $zeta$ is a variance parameter of normal distribution to avoid $p$ converges to $1.0$ by giving very small random noise to oversmoothed features.

---

> ### Author Response · Authors · 2023-11-22
> **Response to Reviewer ntqg - Minor (1), Question (2)**
>
> > The claim “there has been no work on diffusion models for graph representation learning in both feature and structural aspects” is inappropriate because there exist related works such as MoleculeSDE[1],[2].
>
> Our statement "there has been no work on diffusion models for graph representation learning in both feature and structural aspects" holds true without any exaggeration.
>
> To address potential misunderstanding about the meaning of our work, we will provide a clearer explanation of the significance of GDM.
> The motivation behind GDM is that existing graph representation learning models have focused on defining task-dependent / task-specific assumptions for a particular graph learning task. For instance, existing GNNs that mostly rely on message-passing, usually show good performance in node classification tasks while showing limited results in some link prediction tasks. Consequently, there are no powerful models that are capable of solving both link prediction and node classification tasks.
> It is noteworthy that we initially raised the need for this motivation since such motivation has not been raised before.
>
> To mitigate this motivation, we introduce GDM, a diffusion model-based graph representation that is universally applicable to network graph representation learning tasks. Regarding both features and structure of a network graph, the goal of GDM is to learn comprehensive latent representations universally applicable to network graph learning tasks, capturing underlying latent factors from a graph that are inherent in a given task. Leveraging the intuition of diffusion models to capture arbitrary data distributions, GDM learns comprehensive and integrated latent representations crucial for given network graph representation learning tasks (e.g., link prediction) without task-oriented assumptions. Remarkably, it achieves competitive results in both major graph representation learning tasks, link prediction and node classification, without applying task-specific assumptions or biases into a model. As far as our knowledge extends, such research has not been introduced before, highlighting the novelty of our study.
>
> Unlike our GDM, MoleculeSDE[1] and [2] are diffusion models for molecular graph learning which requires readout/pooling to obtain final outputs and diffusion model for graph generation tasks.
> In other words, GDM and [1], [2] target different types of graphs. Molecular graphs are distinctive from network graphs. Molecular graphs have node and edge categorical information, and there are multiple graphs in the dataset, making it possible to define the distribution needed for the diffusion model. This is why diffusion models for graph has been numerously studied. On the contrary, network graphs consist of a single given graph, and sometimes the node's categorical information is incomplete or absent, making it insufficient to define the distribution of the graph. Thus, the diffusion models for graph representation learning has not been investigated thoroughly as molecular graphs.
>
> Indeed, GDM and the reference [1], [2] that the reviewer mentioned are completely different research areas.
> To address the misunderstanding, we will revise the statement it is for *network* graph representation learning.
>
> > What is the relationship between GDM and Digress[1]? The GDM seems to be a specific case of Digress.
>
> GDM is completely different from DiGress (Vignac et al., 2022). It differs in data domain, motivation, and noise source.
>
> Difference from DiGress (Vignac et al., 2022): While GDM is a model for graph representation learning in network-shaped graphs, DiGress is a model for molecular graph generation. In other words, GDM and DiGress target different types of graphs. Molecular graphs have node and edge categorical information, and there are multiple graphs in the dataset, making it easier to define the distribution needed for the diffusion model. However, network graphs consist of a single given graph, and sometimes the node's categorical information is incomplete or absent, making it insufficient to define the distribution of the graph.

---

> ### Author Response · Authors · 2023-11-22
> **Response to Reviewer ntqg - Question (1), Weakness (4)**
>
> > Why the GDM don’t use a powerful GNN as the denoising network? In my understanding, the Loss $L_{diff}$ can be used in any GNN for graph representation learning.
>
> The reason for defining and using our own Denoising network in GDM is as follows: At the convergence state T, node features are strongly oversmoothed and graph structure is highly scarce.
> Since existing GNNs rely on a message-passing mechanism with an adjacency matrix, most of them suffered from oversmoothing problem, indicating limitations in finding the meaningful latent factors from graph data from the convergence state. Therefore, we define the Denoising network with a learnable parameter, the latent Laplacian, to ensure GDM can extract latent information as much as possible.
>
> > Q(1). From the Leaderboards of OGB(https://ogb.stanford.edu/docs/leader_linkprop/), the experimental results of this paper are not very competitive.
> / W(4). The experimental results show the proposed method doesn’t achieve competitive performance in Link prediction (https://ogb.stanford.edu/docs/leader_linkprop/).
>
> The top-ranked performances on the OGB leaderboard are mostly a result of combining various auxiliary techniques such as augmentations or additional plug-in methods.
>
> It is noteworthy that the goal of GDM is to learn latent graph representations universally applicable to network graph learning tasks but capturing underlying latent factors that are intrinsically capable of given network graph learning tasks.
> We compared our model against widely-known effective GNN-based models without such plug-ins, including SEAL which exhibits powerful performance on link prediction tasks among other models in OGB Leaderboard.
>
> Besides, the significance of our work arises from the motivation behind the GDM that existing graph representation learning models are investigated for improving particular graph representation learning tasks, yet they are often vulnerable to other tasks. For instance, in the link prediction task, some existing models assume generalizing some graph heuristics to find missing links. However, those existing models are not capable of learning node embeddings for node classification tasks.
>
> We demonstrate that GDM mitigates its motivation as GDM results in competitive and meaningful performance on both major graph representation learning tasks: link prediction and semi-supervised node classification. Especially, in link prediction tasks, each dataset entails some specifics regarding whether graph structure property is more essential than node embeddings or vice versa. Conventional GNNs show poor performance often since they heavily rely on locality of node embeddings, whereas some competitive models designed for link prediction often fall on a dataset that weights more attention to node embeddings than graph structure properties.
> However, GDM consistently show comparable outputs across all realistic large networks graphs in OGB.
>
> > Some important baselines are missing, such as GIN, on the node classification task.
>
> We indeed know GIN is outstanding in graph representation learning, particularly in graph classification. Since its motivation is to  capture graph isomorphism, ultimately learning small graph representations that are invariant to the same isomorphism, GIN sometimes shows trivial result in network graph representation learning tasks.
>
> As reviewer suggested, we conducted semi-supervised node classification task on OGB-Arxiv. The results are as follows:
>
> |        | **K=1** | **K=5** | **K=10** |
> |----------------|---------|---------|----------|
> | **GIN**        | $5.32\pm{5.92}$  | $29.22\pm{2.28}$ | $34.98\pm{2.43}$  |
> | **GCN**        | $31.69\pm{2.74}$ | $52.97\pm{0.94}$ | $58.39\pm{0.50}$  |
> | **GDM (Ours)** | $38.40\pm{1.64}$ | $57.22\pm{0.85}$ | $60.97\pm{0.40}$  |
>
> Note that our semi-supervised node classification setting is extremely restrained, hence, some GNNs may show low performances if mechanism does not propagate information. We verified that GIN is not suitable for our task.
> Indeed, our experiments did not omit strong baselines.

---

> ### Comment · Reviewer_ntqg · 2023-11-23
>
> Thanks for your response!
> Have you updated the pdf?

---

> ### Author Response · Authors · 2023-11-23
>
> Yes, we have updated the revised paper.
>
> We revised some points the reviewer suggested.
> - More detailed explanation of using Laplacian smoothing as noise source to diffuse and dissipate graph signal (Section 4)
> - Revise some confusing notations, terms, equations into more precise and consistent expressions (Section 1, 4)
> - Clarify motivation and novelty of GDM (Section 1)
> - Clarify GDM is for network graph representation learning (Section1)
> - Derivation of loss function (Appendix)
> - Hyperparameter sensitivity analysis (Appendix)
>
> We hope that our response addresses the reviewer's concern and leads to stronger support. We sincerely appreciate the feedback from the reviewer.

---

> > ### Comment · Reviewer_ntqg · 2023-11-23
> >
> > Thanks for the update.
> >
> > I still have several concerns:
> > 1. What is the difference between a network graph and an ordinary graph (like a biology graph)? why your method cannot be used for other graphs and why other methods like DDPM cannot be directly used in this network graph task?  Gaussian assumption in the diffusion model is widely used in different tasks.
> >
> > 2. The approximate q(At|At−1) ≈ q(At|A0) is weird. Is there a specific expression for the error? Can you give the specific expression for p_\theta(A_{t-1}|At) in the second line of Page 13 and how it leads to the MSE loss?  Further, the next equation of Eq 22 has a similar KL divergence expression with it. Why not use the same methods to get the MSE loss $\\| f-L_{t-1}\\|$.
> >
> > 3. Is the adjacency matrix A and node feature X independent in the network graph? In the citation network, the node feature is usually related to the edge. The decomposition of q(Gt−1|Gt, G0) in Eq(9) requires independence.
> >
> > 3.  In Eq 12, the loss contains the downstream information  L_task involved label.  If there are no labels in the network graphs, how to use your method？
> >
> >
> >
> >
> > Due to my above concerns, I find it necessary to maintain the current score for the time being.

---

> ### Author Response · Authors · 2023-11-23
>
> 1. First, a network graph is graph-structured data that represents interactions or relations between objects or entities, where structure is not formed based on some rules such as chemical graphs. Social network, citation network, interaction network, and even knowledge graph are network graphs.
> The term the reviewer mentioned "ordinary graph" is ambiguous. Biology graph is a kind of network graphs. In our link prediction experiments, OGB-DDI is a biology graph, specifically drug interaction network graph.
> Our method, GDM includes all these graphs which are network graphs.
> The conventional DDPMs models can be directly applied to graph-structured data, when it is for graph generation tasks.
> However, the motivation of our work is to improve graph representation learning, which is very different from graph generation tasks. Graph-structured data is non-Euclidean where Gaussian assumption is not trivial, hence, it needs another desirable noise source.
>
> Therefore, we propose a novel approach more suitable that reflects structural characteristics of graph-structured data, which is Laplacian smoothing is our work.
>
> 2. The approximation is based on $ q(X_{t}|X_{0}) = (I-L)^{t} X_{0} = (I-L) X_{t-1}$ where it can be rewritten as $q(X_{t}|X_{0}) = (I-L)^{t} X_{0} = (I-L) X_{t-1} = q(X_{t}|X_{t-1})$ in the forward process. Since we lift the forward feature process onto graph structure, namely structure reflects features in our model, the forward structure process can also be rewritten as $ q(A_{t}|A_{t-1}) \propto (I-L)X_{t-1} = (I-L)^{t-1}X_{0} = q(A_{t}|A_{0}) $ under mild assumption. In other words, $q(A_{t}|A_{0})$ is not different from $q(A_{t}|A_{t-1})$ as the drop probability of $q(A_{t}|A_{t-1}) $ and $q(A_{t}|A_{0}) $ is not that different.
>
> For divergence term, predicting less dissipated features directly indicates dissipated signal on feature because Laplacian smoothing is computaion. However, structure dissipation cannot be directly deblurred due to uncertainty/stochasticity, predicting the amount of dissipated signals on a structure respect to condition of posterior ($L\_{0}-L\_{t-1}) would ensure closer estimation of dissipation.
>
> 3. Feature matrix $X$ and adjacency matrix $A$ is independent initially. $A$ represents adjacency or connectivity of graph structure and $X$ is a matrix consisting of node features.
> In citation network, each node feature is embedded vectors of each documents. The given feature matrix $X$ is independent form $A$. They got dependencies after the message-passing mechanism updates features.
>
> 4. As this is the first approach of diffusion model for graph representation learning, we focused on supervised or semi-supervised setting. If there is no labels, which means unsupervised setting, according to its generative loss (ELBO), GDM can learn latent representation for graph representation learning tasks since GDM is generative approach for graph representation learning.

---

### Official Review · Reviewer_Uwe6 · 2023-11-01

**Soundness:** 3 good
**Presentation:** 3 good
**Contribution:** 2 fair
**Rating:** 5
**Confidence:** 2

**Summary:**

The paper introduces a Graph Dissipation Model (GDM), an innovative framework designed for both link prediction and node classification tasks in graph-structured data. The novelty lies in a coupled diffusion process that merges structure-based and feature-based diffusion mechanisms. Through exhaustive experiments on multiple datasets from the Open Graph Benchmark (OGB), the authors empirically show that GDM outperforms several state-of-the-art methods across different metrics.

**Strengths:**

Comprehensive Approach - The GDM model is versatile in its application as it targets both link prediction and node classification. This comprehensive scope extends its relevance to a broader set of graph-based tasks, making the paper potentially impactful in the field.

Strong empirical results - The paper takes advantage of the Open Graph Benchmark, a standard and well-regarded set of datasets, providing a robust testing ground for the GDM. Additionally, the authors compare GDM against a wide variety of existing methods, both classical and state-of-the-art, to establish its superiority. Overall, the proposed method performs favorably compared with other baselines.

**Weaknesses:**

Omission of graph generation performance - While the paper innovatively adapts the DDPM to graph-based tasks, it focuses solely on node classification and link prediction for evaluation. The absence of comparative performance analysis on graph generation tasks against existing algorithms leaves an important aspect of its applicability unexplored.

Absence of sensitivity analysis - The model introduces several hyperparameters, including weight tuning parameters and the length of diffusion steps. The paper lacks an examination of how variations in these parameters impact the model's performance, making it difficult to fully justify the model's design choices.

Insufficient theoretical underpinning - Despite presenting a novel methodology, the paper falls short in providing an in-depth theoretical discussion to substantiate its claims. Specifically, it asserts that the model "captures latent factors for any given downstream task," but fails to offer comprehensive evidence or discussion that would bolster such a statement.

**Questions:**

This is a follow up of the weakness one: The paper's title claims "DENOISING GRAPH DISSIPATION MODEL IMPROVES
GRAPH REPRESENTATION LEARNING". Is this claim only valid for the proposed denoising graph dissipation model? Do other DDPM model or more generally other graph generation model help improve graph representation learning? Also, have the authors tried to evaluate the graph generation performance method?

---

> ### Author Response · Authors · 2023-11-21
> **Response to Reviewer Uwe6 - Weakness (1), (2)**
>
> Thank you for dedicating time to review our submission and providing important comments! We are pleased to address your feedback as follows:
>
> > Omission of graph generation performance - While the paper innovatively adapts the DDPM to graph-based tasks, it focuses solely on node classification and link prediction for evaluation. The absence of comparative performance analysis on graph generation tasks against existing algorithms leaves an important aspect of its applicability unexplored.
>
> GDM aims to effectively capture and learn the underlying latent factors of a network graph for graph representation learning tasks. Therefore, to demonstrate its capability in learning the comprehensive representation of network graphs, we conducted link prediction tasks and semi-supervised node classification tasks. However, graph generation tasks aim to generate a graph structure different from the input graph while maintaining certain graph statistics or properties. Since such graph generation tasks differ in nature from the motivation of GDM which is to learn feature-structure integrated representations for network graph learning, graph generation is not a suitable task for validating the effectiveness of GDM in addressing our motivation. Therefore, we conducted experiments on link prediction and node classification which are major graph representation learning tasks.
>
>
> > Absence of sensitivity analysis - The model introduces several hyperparameters, including weight tuning parameters and the length of diffusion steps. The paper lacks an examination of how variations in these parameters impact the model's performance, making it difficult to fully justify the model's design choices.
>
> Thank you for the suggestion. To address your concerns about hyperparameters, we conducted the sensitivity analysis on OGB-Collab. The results are as follows:
>
> | Timestep T  | 6    | 15   | 20   | 30   |  $\lambda$(Recon)  | 0.001 | 0.01  | 0.1   | 1     | $\gamma$(Struc_Diss)  | 0.003 | 0.03  | 0.3   | 1  |  $\beta_{r}$(feat_pred)  | 0.002 | 0.02  | 0.2   | 1     | $\beta_1$(Recon_feat)  | 0.002 | 0.02  | 0.2   | 1     |
> |-----|------|------|------|------|------|-------|-------|-------|-------| ---|-------|-------|-------|-------| ---|-------|-------|-------|-------| ---|-------|-------|-------|-------|
> | **Accuracy** | 53.89| 53.92| 54.01| 53.84| **Accuracy** | 52.28 | 52.54 | 52.80 | 53.89| **Accuracy** | 52.73 | 53.89 | 52.32 | 52.98| **Accuracy** | 52.11 | 53.89 | 52.35 | 52.77| **Accuracy** | 52.95 | 53.44 | 53.89 | 53.36|
>
> As can be seen from the above results, the overall hyperparameters show robust performance.
> We add a more legible version of the hyperparameter analysis in the Appendix.

---

> ### Author Response · Authors · 2023-11-21
> **Response to Reviewer Uwe6 - Weakness (3)**
>
> > Insufficient theoretical underpinning - Despite presenting a novel methodology, the paper falls short in providing an in-depth theoretical discussion to substantiate its claims. Specifically, it asserts that the model "captures latent factors for any given downstream task," but fails to offer comprehensive evidence or discussion that would bolster such a statement.
>
> GDM entails two perspectives: the diffusion model perspective and graph spectral perspective, both of which were discussed in Preliminary (Sec.2) of our work. The goal of GDM is to learn the underlying latent factor (distribution) of the network graph necessary for a given graph representation learning task. However, existing GNNs struggle to capture this sufficiently. For instance, while existing GNNs perform well in node classification tasks, they may fail to find the necessary latent factors when homophily is low or the number of learnable labels is extremely limited. Prior studies have therefore focused on improving task performance by training specialized assumptions for a particular task. We raise the need for capturing graph latent factors for a given graph representation learning task without assumptions dependent on that task.
>
> The diffusion models capture the latent data distributions of images in pixel space that are arbitrary and challenging to capture directly. Therefore, diffusion models utilize the known Gaussian distribution as the source, add randomly sampled noise, and denoise it progressively to learn the arbitrary data distribution. Based on the philosophy of the diffusion model, we introduce GDM for finding the underlying latent factor of a network graph.
> However, unlike images, network graph data do not exist on widely-known space such as a  2-dimensional plane. Therefore, we approach from a graph spectral perspective and define the noise source using Laplacian smoothing. Laplacian smoothing gradually reduces the differences between signals on the graph by decaying frequencies. This is connected to reducing feature/information discrepancy between different nodes from a graph spatial perspective. When this process occurs gradually, it converges into oversmoothing in graph representation learning.
> Defining noise source as Laplacian smoothing to a network graph causes features to mix, which can be interpreted as noising or blurring from the perspective of the diffusion model.
> IHDM (Rissanen et al., 2022) approximates the heat dissipation equation and defines Gaussian blur on an infinite plane. Based on this, when we iteratively apply graph Laplacian smoothing, decays of frequency can be interpreted as information or signal dissipation. In this context, unlike conventional diffusion models or molecular graph diffusion models, our approach does not heavily rely on Markcov chain property, resulting in the capability of learning latent factors by deblurring or directly predicting previous states, as shown in $X_{t} = (I - \alpha {L})^{t} X_{0} = U (I - \alpha \Lambda)^{t} U^{\top} X_{0}$ (Eq.2) in our study.
> Besides, our approach reflects dependencies between data instances (i.e., nodes), considering the structural characteristics of a network graph.
>
> Thank you for pointing out this aspect. We will address this concern in the revised manuscript, and these revisions would enhance the theoretical discussion of our proposed model and provide readers with a more comprehensive understanding of GDM's capabilities.

---

> ### Author Response · Authors · 2023-11-21
> **Response to Reviewer Uwe6 - Questions**
>
> > The paper's title claims "DENOISING GRAPH DISSIPATION MODEL IMPROVES GRAPH REPRESENTATION LEARNING". Is this claim only valid for the proposed denoising graph dissipation model? Do other DDPM model or more generally other graph generation model help improve graph representation learning? Also, have the authors tried to evaluate the graph generation performance method?
>
> This is a crucial point of our contribution! Yes, our claim that GDM improves graph representation learning is valid. Also, other DDPM models or diffusion model-based graph generation models would not effectively improve graph representation learning.
> To emphasize contributions of our work, let us provide a detailed explanation of the motivation behind GDM. The motivation behind GDM is to learn latent graph representations universally applicable to network graph learning tasks, capturing underlying latent factors entailed from a given network graph learning task. We initially raise this motivation, implying the significant contribution of GDM.
> Note that, existing models do not align with our motivation.
>
> However, other DDPM models and diffusion model-based graph generation models introduced so far are designed methods to enhance image generation and graph generation, respectively.  DDPM models may not improve graph representation learning since they do not consider the structural characteristics of a network graph. The essence of graph generation and graph representation learning is completely different. Graph generation tasks aim to generate a graph structure different from the input graph while maintaining certain graph properties, whereas graph representation learning is to learn latent representations from a network graph that are essential for prediction or classification of instances within a given graph.
>
> For the question of graph generation performance, as the same point has been mentioned in Weakness above,
> we respond in Weakness (1).
>
> We hope that our response addresses the reviewer's concern and leads to stronger support. We appreciate your feedback to improve our paper.

---

> > ### Comment · Reviewer_Uwe6 · 2023-11-23
> >
> > I sincerely appreciate the authors' efforts in providing additional sensitivity analysis, which underscores their dedication to enhancing graph representation learning. The evidence put forth about the efficacy of the proposed DDPM-based algorithm is commendable. Nevertheless, the rationale behind how the introduction of a modified DDPM model leads to these improvements remains somewhat obscure. The current explanations, while insightful, seem to require further depth. For instance, claims like 'capturing underlying latent factors' in the paper would greatly benefit from more robust theoretical support or clearer examples. Although the response adequately explains the application of Laplacian smoothing, it doesn't fully bridge the gap in understanding how the proposed methods enhance the capacity to capture latent factors. I am keenly looking forward to a more rigorous theoretical or empirical analysis in the revised version of this paper. However, due to my lingering concerns, I find it necessary to maintain the current score for the time being.

---

> > > ### Author Response · Authors · 2023-11-23
> > >
> > > Thank you for the response.
> > > We attached the revised paper. We add more detailed discussion based on graph spectral in section 4, bridging the gap between our GDM and the intuition of existing DDPMs models. As we leverage Laplacian smoothing to diffuse and dissipate graph signal, it is analogous to diffusion models with coarse-to-fine strategy in image resolution domain.
> > > Since latent factors of a network graph are complex and arbitrary, it is impossible to specify latent factors as known probabilistic distribution families such as Gaussian.
> > > We hope that additional discussion about graph spectra in the revised paper addresses the reviewer's concern and provides some understanding of our proposed model, leading to support.

---

### Official Review · Reviewer_7MVc · 2023-11-01

**Soundness:** 3 good
**Presentation:** 2 fair
**Contribution:** 2 fair
**Rating:** 5
**Confidence:** 3

**Summary:**

The authors proposed a graph denoising diffusion model using Laplacian smoothing and edge deletion as the noise source. Authors claimed their new model achieve better and more general graph representation learning.

**Strengths:**

This is an interesting topic to apply DDPM on graph representation learning. The authors had some good ideas on using Laplacian smoother and a coupled node feature similarity based edge removal schedule to add noises. They claimed this helps learn a more general representation by capturing both the attributes and graph structures.. There are some experiment results to seem to support it.

**Weaknesses:**

The extension of Rissanen et al., 22' work, using Laplacian smoothing for graphs, was natural and even mentioned in the original paper's discussion section. And the claim of *no work on diffusion models for graph representation learning in both feature and structural aspects* feels like an exaggeration. In Vignac et al. 22' (also cited in the manuscript) uses both node features and structural information.

The experiments are not convincing to support authors' claim on the new GDM. Does it learn both feature and structural level information: table 1 only showed it outperforms SEAL on DDI and underperforms on the other three tasks.

**Questions:**

1. The authors need more experiments/analysis to support the claim that their model can learn both features/structural information well.
2. It would be more helpful if the authors can explore a bit more on the spectral meanings of Laplacian smoothing aside from information dissipation...the authors did mention it decays the high-frequency components on the spectral domain. Can we expand this more? Do we gain additional insights from using Laplacian smoothing.
3. I assume GDM was trained on sampled subgraphs (?) but there was no mentioned on how this was done. Does the model only work on smaller graphs?
4. Minior:

    a). In the abstract, *...model leverages Laplacian smoothing and subgraph sampling as a noise source.* What does subgraph sampling mean here? Edge removal?

    b). In the abstract, *...Graph dissipation model that captures latent factors for any given downstream task.* need to tune down.

    c). Some parts of the paper are overly verbose, for example is Corollary 3.1 truly needed?

    d). typos...for example pg5 *graph-strudtured*, pg8, *iffusion*

---

> ### Author Response · Authors · 2023-11-20
> **Response to Reviewer 7MVc - Weakness (1)**
>
> Thank you for dedicating time to review our submission and providing pivotal comments! We are pleased to address your feedback as follows:
>
> > 1. The extension of Rissanen et al., 22' work, using Laplacian smoothing for graphs, was natural and even mentioned in the original paper's discussion section. And the claim of no work on diffusion models for graph representation learning in both feature and structural aspects feels like an exaggeration. In Vignac et al. 22' (also cited in the manuscript) uses both node features and structural information.
>
> Our statement holds true without any exaggeration. Also, the studies the reviewer mentioned are significantly different from GDM.
> To address potential misleading about the meaning of our work, we will provide a clearer explanation of the significance of GDM.
>
> The motivation behind GDM is to learn latent graph representations universally applicable to network graph learning tasks, capturing underlying latent factors inherent in a given network graph representation learning tasks. Existing work has focused on defining task-dependent / task-specific assumptions for a particular graph learning task. For instance, in the link prediction task, some existing models assume generalizing some graph heuristics to find missing links. However, those existing models are not capable of learning node embeddings for node classification tasks. On the other hand, existing GNNs usually show good performance in node classification tasks while showing limitation in link prediction tasks. Hence, there are no powerful models that can solve both link prediction and node classification tasks.
>
> To address this issue, we introduce GDM, a diffusion model-based graph representation learning approach that considers both features and structure of a network graph. Leveraging the capabilities of diffusion models to capture arbitrary data distributions, GDM learns comprehensive and integrated latent representations crucial for given network graph representation learning tasks (e.g., link prediction). Remarkably, it achieves competitive results in both major graph representation learning tasks, link prediction and node classification, without applying task-specific assumptions or biases into a model. As far as our knowledge extends, such research has not been introduced before, highlighting the novelty of our study.
>
> Besides, GDM is completely different from IHDM (Rissanen et al., 2022) and DiGress (Vignac et al., 2022).
> It differs in data domain, motivation, and noise source.
> - Difference from IHDM (Rissanen et al., 2022):
> Unlike GDM, IHDM incorporates the multi-resolution nature of the image domain as an inductive bias to enhance the quality of image generation. Not only is IHDM not in the graph domain, but its motivation is also different as it focuses on image generation. While both GDM and IHDM use smoothing/blurring in the forward process, their purposes and definitions differ. GDM defines the noise source as Laplacian smoothing since nodes, the data instances in the graph domain, have dependencies, necessitating the incorporation of this characteristic as a noise source. Therefore, we define GDM's noise source as Laplacian smoothing, resulting in an oversmoothed final state graph. In contrast, IHDM incorporates the multi-resolution nature of images and uses approximated Gaussian blur in continuous space.
> - Difference from DiGress (Vignac et al., 2022):
> While GDM is a model for graph representation learning in network-shaped graphs, DiGress is a model for molecular graph generation. In other words, GDM and DiGress target different types of graphs. Molecular graphs have node and edge categorical information, and there are multiple graphs in the dataset, making it easier to define the distribution needed for the diffusion model. However, network graphs consist of a single given graph, and sometimes the node's categorical information is incomplete or absent, making it insufficient to define the distribution of the graph.

---

> ### Author Response · Authors · 2023-11-20
> **Resonpose to Reviewer 7MVc - Weakness (2), Question (1), (2)**
>
> > 2. The experiments are not convincing to support authors' claim on the new GDM. Does it learn both feature and structural level information: table 1 only showed it outperforms SEAL on DDI and underperforms on the other three tasks.
>
> The motivation behind GDM is to capture latent factors inherent in network graph representation learning tasks through the intuition of diffusion models, thereby learning integrated graph representations that encompass the feature- and structure-related latent information necessary for a given task. To validate this, we conducted experiments focusing on the Link Prediction task. This is because Link Prediction requires learning not only node embeddings but also structural information [1].
> However, The dependencies on node feature information and graph structural information vary with the dataset. Referring to Table 1 in our work, it implies that OGB-PPA and OGB-DDI place higher importance on structural information and feature information, respectively. OGB-Collab and Citation2 imply the necessity of both types of information for optimal prediction.
>
> Our model, GDM, consistently shows competitive (best or second-best) performance across all four datasets. While SEAL, a model specialized for link prediction, also demonstrates good performance, it shows poor performance in OGB-DDI, compared to GDM. Given that OGB-DDI appears to heavily rely on node embeddings, this result demonstrates that GDM is capable of capturing both latent factors that are informative for solving a given task, unlike other models specialized to specific factors. In other words, GDM’s competitive performance across the four datasets indicates its ability to effectively extract and learn integrated representation about the necessary structure and features for link prediction within a given graph. It is important to note that we did not incorporate task-specific biases into the model. Therefore, we empirically validated GDM has addressed its motivation, demonstrating its ability to capture essential underlying latent factors in a given graph required to handle the link prediction task.
>
> > Q1. The authors need more experiments/analysis to support the claim that their model can learn both features/structural information well.
>
> To demonstrate that our model effectively learns both feature and structural information in network graphs, we conducted link prediction experiments, which are known to require not only node embeddings but also graph structural information. In these experiments, our model shows highly competitive performance, verifying its ability to capture both aspects effectively. Additionally, to demonstrate its ability to learn node embedding, we conducted semi-supervised node classification tasks. GDM outperforms on all datasets. Furthermore,  through the ablation study, we analyzed the validity of whether GDM effectively captures feature and structure latent factors by feature process and structural process.
>
> > Q2. It would be more helpful if the authors can explore a bit more on the spectral meanings of Laplacian smoothing aside from information dissipation...the authors did mention it decays the high-frequency components on the spectral domain. Can we expand this more? Do we gain additional insights from using Laplacian smoothing.
>
> Yes, we would like to provide the explanation of the insight into the decay of high frequency.
>
>  As high frequency gradually decays, the difference between signals will also gradually diminish. In the spatial domain of a graph, it is interpreted as a loss of information regarding the distinct features among nodes. This implies that the amount of lost signal or information varies for each node at each time step, suggesting that our model GDM can learn the latent factors of a given graph by recovering this dissipated signal or information. This aligns with the philosophy and characteristics of diffusion models.
> Additionally, in the real world, there can be noise or missing information (e.g., missing links) in the features or adjacency of a network graph. In other words, observations may not constitute a perfect ground truth. In real-world scenarios, graph representation learning involves learning from a noisy observed graph to approach a more optimal graph representation. From the resolution perspective, our insight is analogous to utilizing a coarse-to-fine strategy to enhance image resolution quality.
>
>
>
> [1] Link prediction based on graph neural networks

---

> ### Author Response · Authors · 2023-11-20
> **Response to Reviewer 7MVc - Question (3), Minor**
>
> > Q3. I assume GDM was trained on sampled subgraphs (?) but there was no mentioned on how this was done. Does the model only work on smaller graphs?
>
> Sorry for the confusion. Indeed, GDN can learn large network graphs. OGB datasets where we conduct experiments are realistic and large-scale graph datasets. Regarding the question about subgraph sampling mentioned in the abstract, subgraph sampling refers to edge removal. We define feature-dependent stochastic sampling to progressively lift feature dissipation into the structure, gradually dropping existing edges from a given graph. Regarding edge sampling, during the implementation of the reverse process, the denoising task is performed on randomly sampled edges from the set of dropped edges instead of all of them for efficiency.
> We will clarify this in the revised manuscript.
>
> > Minor a). In the abstract, ...model leverages Laplacian smoothing and subgraph sampling as a noise source. What does subgraph sampling mean here? Edge removal?
>
> Sorry for the confusion. As the same point has been mentioned in Question (3), we respond with the same explanation in Question (3).
> We will clarify the term in the revised manuscript.
>
> > Minor b). In the abstract, ...Graph dissipation model that captures latent factors for any given downstream task. need to tune down.
>
> Since this point aligns with Weakness (1), we can tell our statement is without exaggeration. We raise the motivation of universally applicable model that learns network graph representation while naturally considering specifics of a given task without injecting task-oriented biases. To address the motivation, we proposed GDM and demonstrated effectiveness in major graph representation learning tasks: link prediction task and semi-supervised node classification task.
>
> However, we will rephrase "for any given downstream task" into "for any given graph representation learning task." to distinguish it from graph generation tasks which is a very different research area.
>
> > Some parts of the paper are overly verbose, for example is Corollary 3.1 truly needed?
>
> Since understanding diffusion models and over-smoothing requires background knowledge, we decided to provide detailed explanations for a more straightforward understanding of intuition behind GDM, which is inseparably related to diffusion models and relation between Laplacian smoothing and over-smoothing.
>
> Thanks to the reviewer, we will fix typos in the revised manuscript.
>
> We hope that our response addresses the reviewer's concern and leads to stronger support.
> We appreciate your feedback to improve our paper.

---

> ### Author Response · Authors · 2023-11-23
>
> We uploaded the revised paper.
> We revised some points the reviewer suggested.
> - More detailed explanation of using Laplacian smoothing as noise source to diffuse and dissipate graph signal (Section 4)
> - Revise some confusing notations, terms, equations into more precise and consistent expressions (Section 1, 4)
> - Clarify motivation and novelty of GDM (Section 1)
> - Clarify GDM is for network graph representation learning (Section1)
> - Derivation of loss function (Appendix)
> - Hyperparameter sensitivity analysis (Appendix)
>
> We hope that our response addresses the reviewer's concern and leads to stronger support.
> We sincerely appreciate the feedback to improve the paper.

---

### Meta-Review · Area_Chair_JYgG · 2023-12-06

**Metareview:**

The paper introduces a Graph Dissipation Model (GDM) for graph representation learning. GDM combines feature-based and structure-based diffusion processes to capture latent factors in graphs, making it suitable for various downstream tasks. It uses Laplacian smoothing for feature diffusion and edge removal for structure diffusion. The paper claims that GDM achieves competitive performance on link prediction and node classification tasks compared to existing graph representation learning methods.

Strengths of the paper:
* The paper addresses both link prediction and node classification, making GDM versatile and relevant for various graph-based tasks.
* GDM introduces innovative techniques like Laplacian smoothing and edge removal as noise sources for diffusion, which can contribute to improved graph representation learning.

Weaknesses of the paper:
* The paper falls short in providing a deep theoretical discussion to support its claims, particularly the assertion that GDM captures universal factors for downstream tasks. More theoretical analysis could strengthen the paper's contributions.
* The paper focuses on node classification and link prediction but does not explore GDM's performance in graph generation tasks. Evaluating GDM in graph generation could provide a more comprehensive understanding of its capabilities.
* The paper introduces various hyperparameters, but there is limited analysis of how different settings impact GDM's performance. A sensitivity analysis would help justify design choices.

**Justification For Why Not Higher Score:**

GDM shows promise in its empirical results. However, it could benefit from more theoretical underpinning, broader task exploration, hyperparameter analysis, and improved presentation.

**Justification For Why Not Lower Score:**

N/A

---

### Decision · Program_Chairs · 2024-01-16

Reject